# Simultaneous trimodal single-cell measurement of transcripts, epitopes, and chromatin accessibility using TEA-seq

**Elliott Swanson[1], Cara Lord[1†], Julian Reading[1], Alexander T Heubeck[1], Palak C Genge[1], Zachary Thomson[1], Morgan DA Weiss[1], Xiao-jun Li[1], Adam K Savage[1], Richard R Green[1,2‡], Troy R Torgerson[1,3], Thomas F Bumol[1], Lucas T Graybuck[1*], Peter J Skene[1*]**

[1]Allen Institute for Immunology, Seattle, United States; [2]Department of Biomedical Informatics and Medical Education (BIME), University of Washington, Seattle, United States; [3]Department of Pediatrics, University of Washington, Seattle, United States

**\*For correspondence:**
lucasg@alleninstitute.org (LTG);
peter.skene@alleninstitute.org (PJS)

**Present address:**
[†]GlaxoSmithKline, Collegeville, United States; [‡]Department of Biomedical Informatics and Medical Education, University of Washington School of Medicine, Seattle, United States

**Abstract** Single-cell measurements of cellular characteristics have been instrumental in understanding the heterogeneous pathways that drive differentiation, cellular responses to signals, and human disease. Recent advances have allowed paired capture of protein abundance and transcriptomic state, but a lack of epigenetic information in these assays has left a missing link to gene regulation. Using the heterogeneous mixture of cells in human peripheral blood as a test case, we developed a novel scATAC-seq workflow that increases signal-to-noise and allows paired measurement of cell surface markers and chromatin accessibility: integrated cellular indexing of chromatin landscape and epitopes, called ICICLE-seq. We extended this approach using a droplet-based multiomics platform to develop a trimodal assay that simultaneously measures transcriptomics (scRNA-seq), epitopes, and chromatin accessibility (scATAC-seq) from thousands of single cells, which we term TEA-seq. Together, these multimodal single-cell assays provide a novel toolkit to identify type-specific gene regulation and expression grounded in phenotypically defined cell types.

## Introduction

Peripheral blood mononuclear cells (PBMCs) purified using gradient centrifugation are a major source of clinically relevant cells for the study of human immune health and disease (**Böyum, 1968**). Like most other human tissues, PBMCs are a complex, heterogeneous mixture of cell types derived from common stem cell progenitors (**Laurenti and Göttgens, 2018**). Despite the genome being mostly invariant between different PBMC cell types, each immune cell type performs an important and distinct function. Understanding the genomic regulatory landscape that controls lineage specification, cellular maturation, activation state, and functional diversity in response to intra- and extracellular signals is key to understanding the immune system in both health and disease (**Satpathy et al., 2019**; **Wang et al., 2020**; **Zheng et al., 2020**).

Recent improvements in single-cell genomic methods have enabled profiling of the regulatory chromatin landscape of complex cell-type mixtures. In particular, droplet-based single-nucleus or single-cell assays for transposase-accessible chromatin (snATAC-seq, scATAC-seq, dscATAC-seq, mtscATAC-seq) allow profiling of open chromatin at single-cell resolution (**Buenrostro et al., 2015**; **Lareau et al., 2019**). Promising new methods have combined scATAC-seq with simultaneous measurement of nuclear mRNAs (e.g., sci-CAR, **Cao et al., 2018**; SNARE-seq, **Chen et al., 2019**; SHARE-seq, **Ma et al., 2020**) or in combination with cell surface epitopes (ASAP-seq,

*Mimitou et al., 2020*). However, a unifying approach for all three modalities that can be applied to highly specified functional immune cell types has yet to emerge in the landscape of single-cell methods. We systematically tested whole cell and nuclear purification and preparation methods for PBMCs to overcome limitations that restricted previous assays to measurement of only nuclear components (ATAC and nuclear RNAs) or proteins on the cell surface. We found that intact, permeabilized cells perform extremely well for scATAC-seq, exceeding conventional scATAC-seq on nuclei by some measures (*Figure 1b*). This insight enables a new protocol analogous to Cellular Indexing of Transcriptomes and Epitopes (CITE-seq; *Stoeckius et al., 2017*) to measure both surface protein abundance and chromatin accessibility: integrated cellular indexing of chromatin landscape and epitopes (ICICLE-seq, *Figures 1a* and 3). Finally, we demonstrate that our optimized permeable cell approach can be combined with a droplet-based multiomics platform to enable the simultaneous measurement of three different molecular compartments of the cell: mRNA (by scRNA-seq), protein (using oligo-tagged antibodies), and DNA (by scATAC-seq), which we term TEA-seq after Transcription, Epitopes, and Accessibility (Figure 4). These assays enable a new, more unified view into the molecular underpinnings of gene regulation and expression at the single-cell level.

## Results

### Optimization of single-nucleus and single-cell ATAC-seq of PBMCs

As a baseline for optimization and cell surface retention, we performed snATAC-seq as recommended by 10x Genomics, with a protocol based on the Omni-ATAC workflow (*Corces et al., 2017*). This single-nucleus assay utilizes a combination of hypotonic lysis, detergents, and a saponin to isolate nuclei without retaining mitochondrial DNA. After performing snATAC-seq using this method, sequencing, and tabulating data quality metrics (Materials and methods), we identified two major populations of cell barcodes (*Figure 1b*, left panel): (1) a large number of barcodes, shown in gray, that have a low number of unique fragments and a low fraction of reads in peaks (FRIP). These barcodes contain little useful information but consume 80% of total sequenced reads (*Figure 1e*, non-cell barcodes) at a sequencing depth of 200 million reads per library (20,000 reads per expected barcode). (2) Barcodes with higher quality as measured by FRIP (red points) that contain enough information to attempt downstream analysis.

The loss of 80% of sequenced reads to non-cell barcodes is costly. Previous studies of scRNA-seq data have shown that cellular lysis can release ambient RNA that increases the abundance of low-quality barcodes and contaminates droplets, yielding barcodes with both cellular and ambient RNAs that reduces the accuracy of the transcriptional readout (*Marquina-Sanchez et al., 2020*). We reasoned that nuclear isolation protocols may cause the release of ambient DNA, causing a similar effect in scATAC-seq datasets. Optimization of nuclear lysis protocols, especially changing to less stringent detergents, provided increased FRIP and decreased non-cell barcodes (*Figure 1—figure supplement 1*, *Figure 1—figure supplement 2a*). Hypotonic lysis conditions used in these protocols may also be a biophysical stressor to the native chromatin state, as previously observed (*Lima et al., 2018*). To reduce perturbation of chromatin and retain the cell surface for multimodal assays, we performed cell membrane permeabilization under isotonic conditions to allow access to the nuclear DNA without isolating nuclei through hypotonic lysis. The saponin digitonin was used to cause concentration-dependent selective permeabilization of cholesterol-containing membranes while leaving inner mitochondrial membranes intact, preventing high levels of Tn5 transposition in mitochondrial DNA (*Adam et al., 1990*; *Colbeau et al., 1971*). Digitonin has previously been used for ATAC-seq assays under hypotonic conditions in Fast-ATAC (*Corces et al., 2016*) and plate-based scATAC-seq (*Chen et al., 2018b*) protocols. Permeabilization of intact cells under isotonic conditions greatly reduced the number of non-cell barcodes and their contribution to sequencing libraries (*Figure 1—figure supplement 2b*).

### Removal of neutrophils greatly increases PBMC scATAC-seq quality

We observed that PBMCs purified by leukapheresis rather than Ficoll gradient centrifugation had consistently higher FRIP scores and fewer non-cell barcodes (*Figure 1—figure supplement 2c*). A major difference between Ficoll-purified PBMCs and leukapheresis-purified PBMCs was the presence of residual neutrophils in Ficoll-purified samples. We tested removal of dead cells and debris with

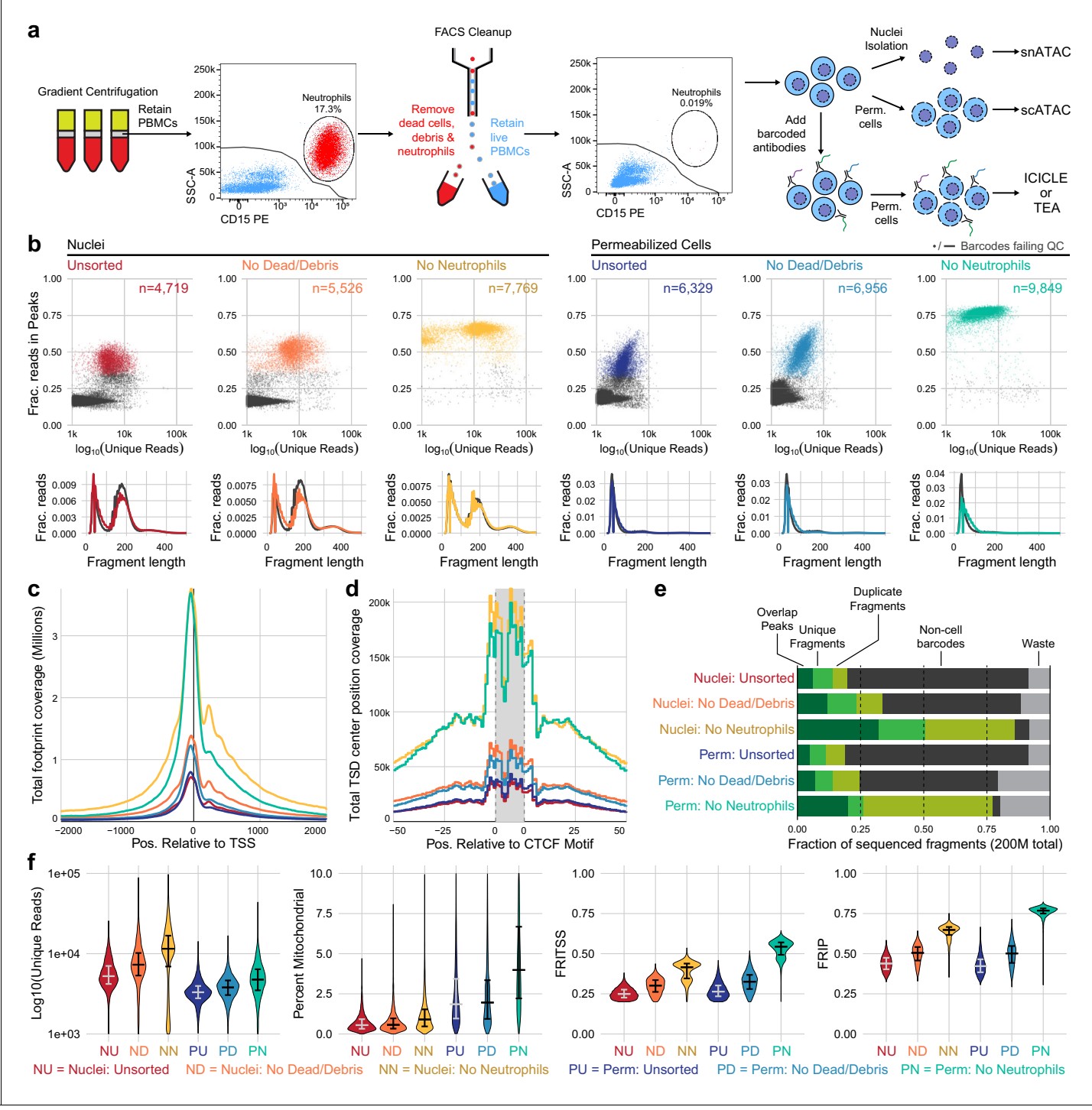

**Figure 1.** Improvements to scATAC-seq methods to enable permeabilized cell profiling. (**a**) Schematic overview of major steps in snATAC, scATAC, and ICICLE-seq methods. (**b**) Comparison of quality control characteristics of ATAC-seq libraries generated from nuclei isolation and permeabilized cells, with and without fluorescence-activated cell sorting. Top panels show signal-to-noise as assessed by fraction of reads in peaks on the y-axis and quantity of unique fragments per cell barcode on the x-axis. Lower panels display fragment length distributions obtained from paired-end sequencing of ATAC libraries. Colored lines represent barcodes that pass QC filters; gray lines represent barcodes failing QC (non-cell barcodes). All libraries were equally downsampled to 200 million total sequenced reads for comparison. Colors in (**b**) are reused in remaining panels. (**c**) Total coverage of Tn5 footprints summed across all transcription start sites (TSS). Tn5 footprints are 29 bp regions comprising the 9 bp target-site duplication (TSD) and 10 bp on either side, which represent accessible chromatin for each transposition event. (**d**) Total coverage of TSD centers summed over a set of 100,000 genomic CTCF motifs found in previously published DNase-hypersensitive sites (*Meuleman et al., 2020*). TSD centers are obtained by shifting +4 and

*Figure 1 continued on next page*

*Figure 1 continued*

−5 bp from the 5′ and 3′ ends of uniquely aligned fragments, respectively. (**e**) Barplot representations of the fraction of total aligned reads in various QC categories. Fragments overlapping a previously published peak set for peripheral blood mononuclear cell dscATAC-seq (*Lareau et al., 2019*) are in the 'Overlap Peaks' category. Unique fragments are the remaining uniquely aligned fragments that do not overlap peak regions. 'Waste' reads were not aligned or were assigned to cell barcodes with fewer than 1000 total reads. (**f**) Violin plots showing distributions of QC metrics. Median (wide bar) and 25th and 75th quantiles (whiskers and narrow bars) are overlaid on violin plots. Median values are also in *Table 1*. Note that the y-axis of the first panel is on a logarithmic scale; remaining panels are linear. snATAC: single-nucleus assays for transposase-accessible chromatin; scATAC: single-cell assays for transposase-accessible chromatin; ICICLE-seq: integrated cellular indexing of chromatin landscape and epitopes.

The online version of this article includes the following source data and figure supplement(s) for figure 1:

**Source data 1.** Single cell metadata and QC metrics for scATAC-seq experiments .
**Source data 2.** Fragment size distribution data .
**Source data 3.** TSS Footprint pileups.
**Source data 4.** CTCF Tn5 target site duplication center pileups.
**Source data 5.** Fraction of reads per alignment category.
**Figure supplement 1.** Nuclei were isolated from peripheral blood mononuclear cells using two buffer compositions and varying detergent concentrations.
**Figure supplement 2.** Quality control plots for single-nucleus assays for transposase-accessible chromatin (snATAC-seq) and single-cell assays for transposase-accessible chromatin ( scATAC-seq) experimental conditions.
**Figure supplement 3.** Sequencing depth calculation and projection.
**Figure supplement 4.** Flow cytometry gating to assess neutrophil depletion.
**Figure supplement 5.** Gating strategy for the eight-color flow cytometry panel used to evaluate anti-CD15 bead-based neutrophil removal.
**Figure supplement 6.** Examination of transcription start site-proximal fragment positions across scATAC methods.

and without removal of neutrophils using fluorescence-activated cell sorting (FACS) from PBMC samples with high neutrophil content (*Figure 1a, Figure 1—figure supplement 4*). When applied to either nuclei (*Figure 1b*, left panels) or permeabilized cells (*Figure 1b*, right panels), there was a large increase in FRIP and reduction in non-cell barcodes in our scATAC-seq libraries (*Figure 1e*). Removal of neutrophils did not have an adverse effect on leukapheresis-purified PBMC data (*Figure 1—figure supplement 2c*, right panel). Depletion of neutrophils using anti-CD15 magnetic beads also improved data quality (*Figure 1—figure supplement 2d*, *Figure 1—figure supplement 5*), though not to the same extent as FACS-based depletion. Staining and flow cytometry using an eight-antibody panel (*Supplementary file 1*) on Ficoll- and leukapheresis-purified PBMCs showed minimal effect of the magnetic bead treatment on non-neutrophil cell-type abundance (*Supplementary file 2*).

## Comparing single-cell and single-nucleus ATAC characteristics

We assessed the quantitative and qualitative differences between nuclei and permeabilized cell protocols with and without sorting by performing both protocols on a single set of input cells. To fairly compare data quality across methods, we equally downsampled raw data at the level of raw sequenced reads per well for each sample that we compare directly (see Materials and methods for details). We then utilized a uniform set of transcription start site (TSS) regions (TSS ± 2 kb) and a previously published PBMC peak set (*Lareau et al., 2019*, Materials and methods) as reference regions for computation of fraction of reads in TSS (FRITSS) and FRIP, respectively. Permeabilized cells yielded many more high-quality cell barcodes than nuclear preps using equal loading of cells or nuclei (15,000 loaded, expected 10,000 captured, *Table 1*). scATAC-seq libraries prepared from nuclei contained many more reads originating from nucleosomal DNA fragments (*Figure 1b*, lower panels), and non-cell barcodes from nuclei (gray lines) contained more of these fragments than cell barcodes. Thus, an overabundance of mononucleosomal fragments may indicate non-cell fragment contamination. Libraries from permeabilized cells consisted almost entirely of short fragments, suggesting that permeabilization under isotonic conditions did not loosen or release native chromatin structure at the time of tagmentation (*Figure 1b*, lower panels). Previous bulk ATAC-seq studies have shown that differing nuclear isolation protocols lead to varying amounts of mononucleosomal fragments (*Li et al., 2019a*). In agreement with in vitro experiments studying the effects of low salt on nucleosomal arrays (*Allahverdi et al., 2015*), this further suggests that hypotonic lysis leads to alteration of chromatin structure, raising the possibility of artifactual measurements of accessibility in nuclei-based ATAC-seq. To assess the effect that this difference has on the data obtained by each

**Table 1.** QC metrics summary for experiments displayed in *Figure 1*.

| Source type | FACS depletion | N pass QC | Median N fragments | Median mitochondrial reads | Median unique | Median in TSS | Median in peaks |
|---|---|---|---|---|---|---|---|
| Nuclei | Unsorted | 4719 | 7344 | 43 \| 0.6% | 5247 \| 71.6% | 1332 \| 24.9% | 2306 \| 43.9% |
| Nuclei | Dead/debris | 5526 | 10,526 | 59 \| 0.6% | 7284 \| 69.4% | 2186.5 \| 30.1% | 3647 \| 50.5% |
| Nuclei | Dead/debris/neutrophils | 7769 | 19,972 | 136 \| 0.9% | 11,528 \| 59.1% | 4846 \| 41.5% | 7503 \| 64.8% |
| Permeabilized cells | Unsorted | 6329 | 5541 | 100 \| 1.9% | 3308 \| 59.8% | 871 \| 26.4% | 1390 \| 42.2% |
| | Dead/debris | 6956 | 6733.5 | 120 \| 2.0% | 3795.5 \| 56.8% | 1219.5 \| 32.6% | 1874 \| 50.2% |
| | Dead/debris/neutrophils | 9849 | 14,069 | 514 \| 4.0% | 4756 \| 34.3% | 2536 \| 54.4% | 3650 \| 76.8% |

For all metrics to the right of median N fragments, both the absolute number and a percentage are provided. Median % mitochondrial and median % unique were calculated as a fraction of total fragments; % in TSS and % in peaks were calculated as a fraction of unique fragments. TSS: transcription start sites; FACS: fluorescence-activated cell sorting.

method, we overlaid Tn5 footprints near TSS ( *Figure 1c*) and CTCF transcription factor binding sites (TFBS, *Figure 1d*). The signal at TSS was retained in permeabilized cells, but positions flanking the TSS (occupied by neighboring nucleosomes) had reduced signal compared to isolated nuclei (examined in detail in *Figure 1—figure supplement 6*). At CTCF motifs, we observed nearly identical patterns of accessibility in both nuclei and permeabilized cells, suggesting that scATAC-seq signal at regulatory TFBS is retained in permeabilized cells. Neutrophil and dead cell removal improved the quality of nuclear scATAC-seq libraries, which yielded the highest number of unique fragments, reads in TSS, and reads in peaks, though at the cost of fewer high-quality barcodes captured when compared to permeabilized cells (*Table 1*). Overall, permeabilized intact cells obtained by FACS had the highest FRIP and FRITSS scores, fewest non-cell barcodes, and greatest cell capture efficiency with only a modest increase in mitochondrial reads (*Table 1*, *Figure 1e, f*).

## Improved label transfer and differential analysis

We next examined the effect of methodological differences on downstream biological analyses (*Figure 2*). Removal of neutrophils greatly improved the ability to separate various cell types in uniform manifold approximation and projection (UMAP) projections of both nuclei and cells (*Figure 2a, b*). To provide ground truth for label transfer, we performed flow cytometry on an aliquot of the same PBMC sample used for scATAC-seq above. A panel of 25 antibodies (*Supplementary file 3*) was used to determine the proportion of each of the 12 cell types used to label the scATAC-seq cells in the PBMC sample (*Figure 2—figure supplement 1* and *Supplementary file 4*). Label transfer was performed using the ArchR package (*Granja et al., 2020*) to generate gene scores that enabled label transfer from a reference scRNA-seq dataset using the method provided in the Seurat package (*Stuart et al., 2019*) (Materials and methods). Using these tools, removal of neutrophils improved label transfer scores, and permeabilized cells yielded more cells with high label transfer scores than nuclei-based approaches (*Figure 2b, c*). In addition, permeabilized cells provided labels most similar to the cell-type proportions identified by flow cytometry (*Figure 2d*), with identification of CD8 effector cells only observed in scATAC-seq with permeabilized cells. All methods yielded fewer CD16+ monocytes than observed by flow cytometry, suggesting that CD16+ monocytes may be lost during scATAC-seq using either nuclei or permeabilized cells, or that label transfer methods were not conducive to identifying this cell type (*Figure 2d*). After labeling cell types, we used ArchR to call peaks for each cell type and perform pairwise tests of differential accessibility between each pair of cell types (*Figure 2—figure supplement 2a*). We found many more differentially accessible sites in both cells and nuclei after removal of neutrophils. Differential accessibility was also used to identify differentially enriched TFBS motifs in each cell type (*Figure 2—figure supplement 2b*). Without neutrophil removal (nuclei unsorted, top panel), we were unable to identify significantly enriched motifs in B cells and NK cells that were readily apparent in data from clean nuclei or permeabilized cells (bottom two panels). Together, these results demonstrate that neutrophil removal and the use of permeabilized cells allow for identification of specific cell types and TFBS motifs that are involved in regulation of gene expression.

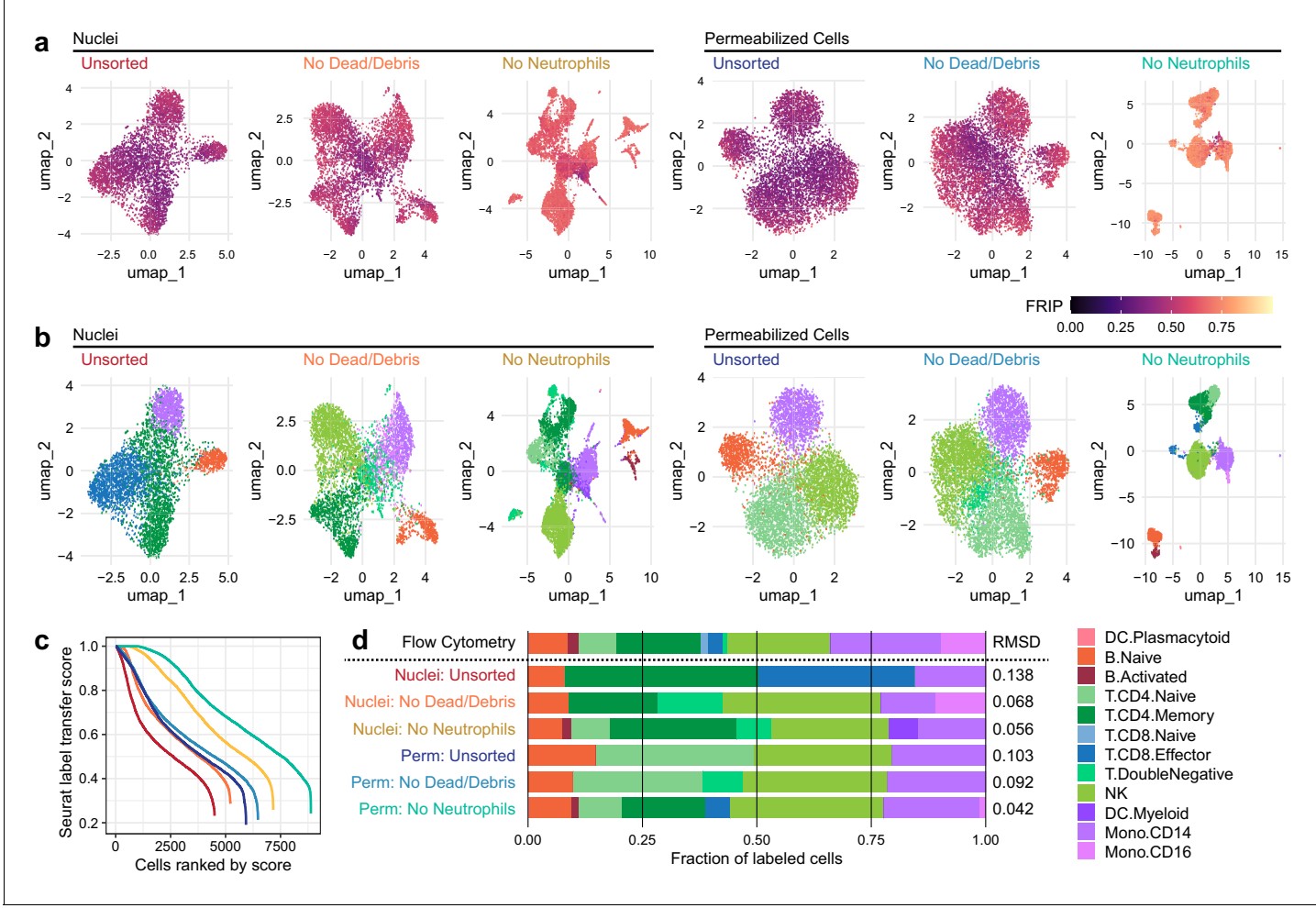

**Figure 2.** Improvements to 2D projection and label transfer for scATAC-seq data. (**a**) Uniform manifold approximation and projection (UMAP) projection plots for corresponding datasets in *Figure 1*. Points are colored based on a common scale of fraction of reads in peaks, bottom right. The number of cells in each panel is displayed in *Figure 1b*. (**b**) UMAP projection plots colored based on cell type obtained by label transfer from scRNA-seq (Materials and methods). Colors for cell types are below, to the right. (**c**) To visualize the number and quality of transferred labels, we ranked all cells based on the Seurat label transfer score obtained from label transfer results and plotted lines through the score (y-axis) vs. rank (x-axis) values. (**d**) Barplot showing the fraction of cells in each dataset that were assigned each cell-type label. The top row shows cell-type proportions for the same peripheral blood mononuclear cell sample obtained by 25-color immunotyping flow cytometry (Materials and methods, *Supplementary file 3*, and *Figure 2—figure supplement 1*). Root-mean-square deviation values were computed by comparison of labeled cell-type proportions to values derived from flow cytometry (*Supplementary file 4*). Colors for cell types are to the right of the barplot.

The online version of this article includes the following source data and figure supplement(s) for figure 2:

**Source data 1.** Single cell UMAP coordinates and labeling scores.
**Source data 2.** Fractions of cells assigned to each type by flow cytometry and scATAC-seq.
**Figure supplement 1.** Flow cytometry gating used to classify and quantify cell-type abundance.
**Figure supplement 2.** Improved diffential peak and motif detection.

## Joint measurement of accessibility and epitopes with ICICLE-seq

Under standard scATAC-seq protocols, removal of the cell membrane severs the connection between the cell surface and the chromatin state of cells. To test our ability to measure cell surface proteins and chromatin state simultaneously on permeabilized cells, we modified our optimized permeabilized cell scATAC-seq methodology to incorporate measurements using commercially available barcoded antibody reagents, and we term this new method ICICLE-seq (*Figure 3* and Materials and methods). The ICICLE-seq protocol utilizes a custom Tn5 transposome complex with capture sequences compatible with the 10x Genomics 3′ scRNA-seq gel bead capture

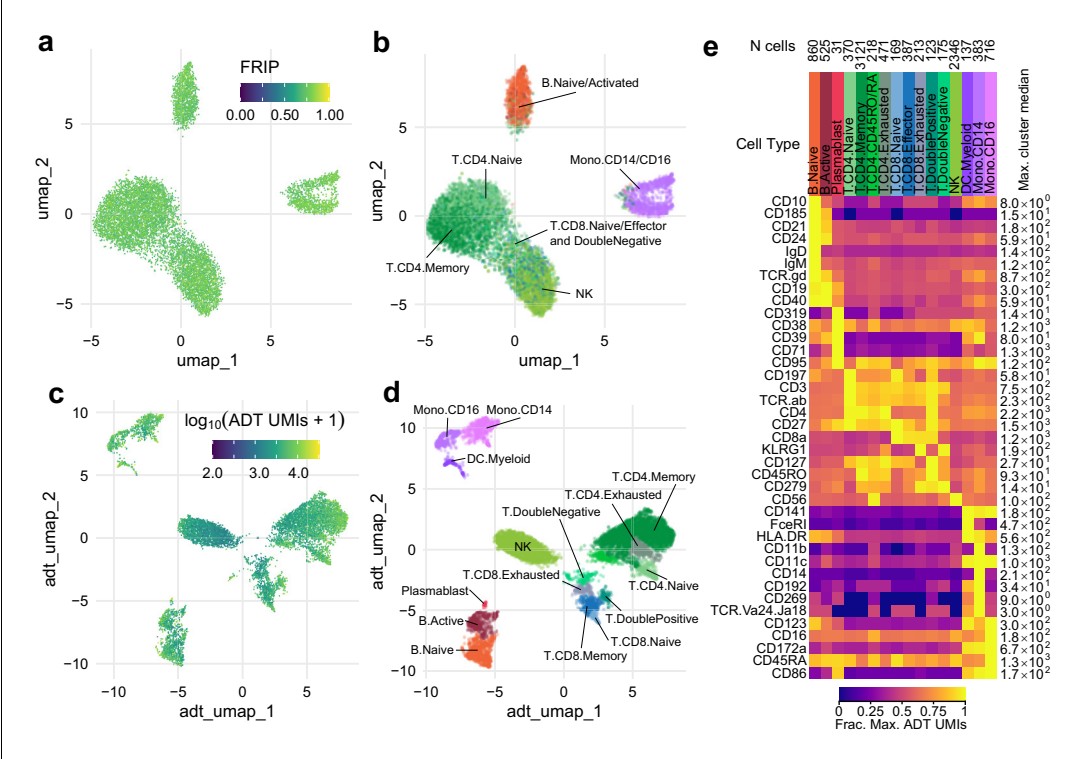

**Figure 3.** Simultaneous profiling of chromatin accessibility and cell surface epitopes. (**a**) Uniform manifold approximation and projection (UMAP) projection plot of integrated cellular indexing of chromatin landscape and epitopes (ICICLE-seq) cells based on single-cell assays for transposase-accessible chromatin (scATAC-seq) data. Cells are colored based on fraction of reads in peaks. n = 10,227 cells passing QC criteria are displayed. (**b**) UMAP projection of scATAC-seq data, as in (**a**). Cells are colored based on cell-type labels obtained by ArchR label transfer (Materials and methods). (**c**) UMAP projection plot of ICICLE-seq cells based on antibody-derived tag (ADT) data. Cells are colored according to the total number of unique molecule indexes across all markers. (**d**) UMAP projection based on ADT data, as in (**c**), colored according to cell-type labels derived from marker expression (Materials and methods). (**e**) Heatmap of median ADT count values for each marker in each cell type labeled in (**d**). Values are separately scaled in each row between zero and the maximum value (right column) for each marker.

The online version of this article includes the following source data and figure supplement(s) for figure 3:

**Source data 1.** Cell type labels and UMAP coordinates for ICICLE-seq cells.

**Source data 2.** ADT count data for ICICLE-seq.

**Figure supplement 1.** Sequence-level integrated cellular indexing of chromatin landscape and epitopes design and workflow.

**Figure supplement 2.** Quality metrics and epitope detection for ICICLE-seq.

reaction for simultaneous capture of ATAC fragments and polyadenylated antibody barcode sequences (*Figure 3—figure supplement 1* and *Supplementary file 5*). Antibody-derived tags (ADTs) from oligo-antibody conjugates and ATAC-seq fragments can then be selectively amplified by PCR to generate separate libraries for sequencing (*Figure 3—figure supplement 1*). Due to the nature of fragment capture in this system, we obtain both a cell barcode and a single-end scATAC-seq read from the two ends of the paired-end sequencing reaction. We performed ICICLE-seq on a leukapheresis-purified PBMC sample using a 46-antibody panel (*Supplementary file 6*) and were able to obtain 10,227 single cells with both scATAC-seq and ADT data from three capture wells that passed adjusted QC criteria: >500 unique ATAC fragments (median = 761), FRIP >0.65 (median = 0.725). Cells passing ATAC QC had a median of 3871 ADT unique molecule indexes (UMIs) per cell (*Figure 3—figure supplement 2b*). UMAP projection and ATAC label transfer on ICICLE-seq data had resolution similar to scATAC-seq on intact permeabilized cells after dead cell and debris removal (*Figure 3b*). However, the data quality was limited by the poly-A-based capture method and the single-end readout of transposed fragments. Nonetheless, the ICICLE-seq results demonstrated that permeabilized cells enabled simultaneous capture of chromatin accessibility in the nucleus and high-quality cell surface epitope quantification. We were able to leverage the additional ADT data to

cluster and identify cell types based on their cell surface antigens (*Figure 3c*) UMAP based on ADT data, and Jaccard–Louvain clustering allowed identification of cell-type-specific clusters (*Figure 3d*) based on clear association of cell-type-specific markers with clusters (*Figure 3e, Figure 3—figure supplement 2c*). Thus, ICICLE-seq provided a key proof of principle for directly linked capture of functional cell types and chromatin accessibility.

## Trimodal measurement of transcripts, epitopes, and accessibility with TEA-seq

With the release of a commercially available platform for simultaneous capture of RNA-seq and ATAC-seq from single nuclei, we reasoned that permeabilized cells could be used to perform simultaneous capture of three major molecular compartments: DNA can be captured using scATAC-seq, RNA could be captured using scRNA-seq, and protein epitope abundance can be captured using polyadenylated antibody barcodes, which we term TEA-seq after Transcripts, Epitopes, and Accessibility. After trials and optimization of key steps, we were able to obtain libraries on the 10x Genomics Multiome ATAC plus Gene Expression platform that combined all three of these measurements for thousands of single cells (*Figure 4a*) using a panel of 46 oligo-tagged antibodies (Supp Table 6). After initial data processing for cells loaded into four wells, we identified 29,264 cell barcodes that passed the QC criteria for scATAC-seq described above and had >2,500 unique ATAC-seq fragments (median = 8762 unique ATAC fragments, *Figure 4b, d*) and had >500 genes detected by scRNA-seq (median = 2399 RNA UMIs; median = 1,249 genes detected, *Figure 4b*), and >500 ADT UMIs (median = 1171 ADT UMIs, *Figure 4c*). Additional QC metrics are provided in *Figure 4—figure supplement 1*.

With the addition of scRNA-seq data, we were able to perform cell-type label transfer from RNA to RNA using Seurat label transfer (*Stuart et al., 2019*) rather than cross-modality transfer from RNA to ATAC (Materials and methods). For each modality, we were able to perform dimensionality reduction and UMAP projection to get separate views of the relationships between cells (*Figure 4e*), as for ICICLE-seq, above (*Figure 3a, b*). However, these views do not leverage the strength of linked multimodal measurements. To begin to take advantage of our tri-modal measurements, we extended a method recently described by Hao and Hao et al. for paired weighted nearest-neighbors (WNN) analysis (*Hao et al., 2020*) to allow an arbitrary number of simultaneously measured modalities to contribute to the WNN network (Materials and methods). After applying this three-way WNN, we could generate a UMAP embedding with contributions from all three simultaneously measured modalities (*Figure 4f*), which enhanced cell-type separation. We found that marker expression is consistent across all three modalities for some markers, such as CD8A, though we found that consistency across all modalities is not a universal characteristic of functional markers (*Figure 4—figure supplement 2a*). In agreement with observations by Hao and Hao et al. on two-modality data, the weights contributed by different modalities to the WNN analyses vary by cell type (*Figure 4—figure supplement 2b*).

## Expanding cis-regulatory module identification with TEA-seq

Identification of putative cis-regulatory modules (CRMs) that control the functional state of cells is a distinct advantage of directly linked measurements of chromatin accessibility and gene expression (*Cao et al., 2018*; *Ma et al., 2020*). This linkage is dependent on reliable detection of gene (or protein) expression, which is greatly enhanced by the high sensitivity of epitope detection in methods like CITE-seq (*Hao et al., 2020*; *Stoeckius et al., 2017*), and now TEA-seq. Because TEA-seq allows direct linkage of chromatin accessibility to both gene expression and protein detection, we used these complementary modalities to investigate peak-to-gene (and peak-to-protein) correlations among the proteins we measured in our TEA-seq epitope panel. We leveraged the analytical capabilities of the ArchR package to compute peak-to-gene correlation (*Granja et al., 2020*), then substituted the more sensitive measurements afforded by ADT counts to compute peak-to-protein correlations. In cases where both gene and protein expression were reliably detected, these comparisons were confirmatory – many of the same putative CRMs that were identified in each comparison (*Figure 4—figure supplement 3*). For some targets, however, more sensitive protein abundance measures allowed detection of many additional correlated CRMs. We present two examples – one of CCR2/CD192 (*Figure 4g, h*), a chemokine receptor expressed on monocytes (*Tsou et al., 2007*),

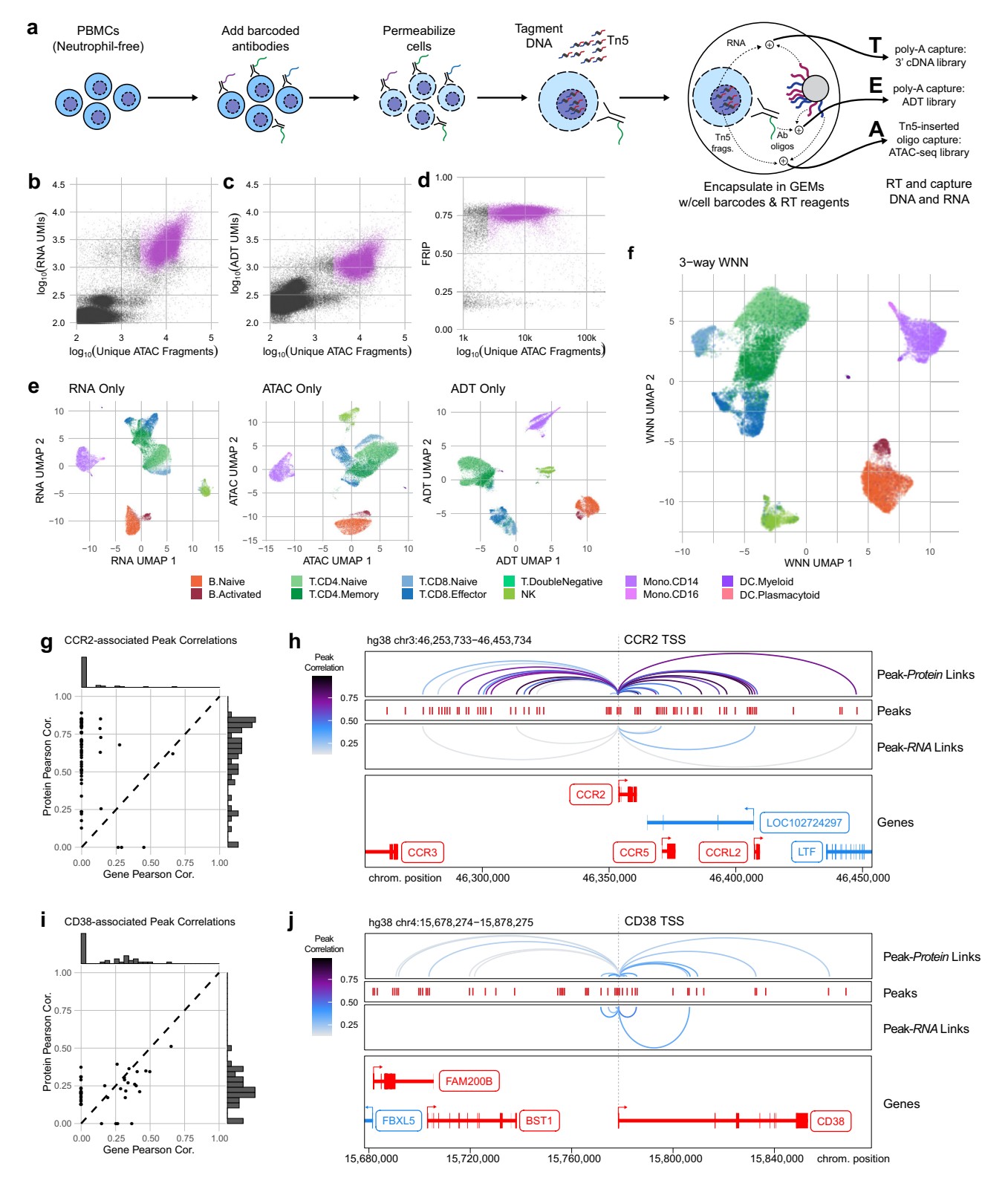

**Figure 4.** Trimodal measurement of transcription, epitopes, and accessibility. (**a**) Workflow diagram for the major steps in transcription, epitopes, and accessibility (TEA-seq). (**b**) Scatterplot comparing unique single-cell assays for transposase-accessible chromatin (scATAC-seq) fragments and scRNA-seq unique molecule indexes (UMIs) for each TEA-seq cell barcode. In (**a**, **b**, **d**, and **e**), n = 227,390 barcodes are displayed in total; 29,264 passing QC criteria are represented by purple points. (**c**) Scatterplot comparing unique scATAC-seq fragments and antibody-derived tags UMIs for each cell

*Figure 4 continued on next page*

*Figure 4 continued*

barcode. (**d**) scATAC-seq QC scatterplot comparing unique scATAC-seq fragments and fraction of reads in peaks scores for each cell barcode. n = 34,757 total cells are displayed (those with >1,000 unique ATAC fragments); 29,264 passing QC criteria are represented by purple points. (**e**) Uniform manifold approximation and projection (UMAP) projections generated using each of the three modalities separately. Only cells passing QC (n = 7,939 barcodes) are presented in (**e, f**). (**f**) A joint UMAP projection generated using three-way weighted nearest neighbors that leverages all three of the measured modalities. (**g**) Scatterplot showing the peak-to-RNA correlations (x-axis) and peak-to-protein (y-axis) correlation values for each peak (points) that was found to be correlated with the CCR2 gene or CD192 antibody in TEA-seq data. Histograms at the margins show the distribution of scores for RNA correlations (top) and protein correlations (right). Dashed line shows 1:1 correspondence. Peaks not found to be correlated in each method were assigned a score of 0. (**h**) Genome tracks showing links between peaks (red hashes) and the CCR2 gene based on protein expression (above peaks) and gene expression (below peaks). Correlations are represented by arcs colored based on the correlation score (color scale for both panels to the left). The bottom panel shows the gene neighborhood around CCR2. All coordinates are from the Hg38 genome assembly. (**i**), as in (**g**), for the CD38 gene and correlated peaks. (**j**), as in (**h**), for the CD38 gene locus.

The online version of this article includes the following source data and figure supplement(s) for figure 4:

**Source data 1.** Single cell quality metrics for TEA-seq samples.
**Source data 2.** Cell type labels and UMAP coordinates for TEA-seq samples.
**Source data 3.** Peak to gene and peak to protein link correlations.
**Figure supplement 1.** Quasirandom-jittered plots (jittered only on x-axis) showing various QC metrics from transcription, epitopes, and accessibility (TEA-seq) cells that passed all QC criteria (Materials and methods).
**Figure supplement 2.** Cell-type marker expression and modality weights.
**Figure supplement 3.** Scatterplot showing the peak-to-RNA correlations (x-axis) and peak-to-protein (y-axis) correlation values for each peak (points) that was found to be correlated with the genes in our antibody-derived tags antibody panel used for transcription, epitopes, and accessibility (TEA-seq) experiments.
**Figure supplement 4.** Direct comparisons of TEA-seq, scATAC-seq, CITE-seq, and 10x Multiome.

and CD38 (*Figure 4i, j*), a glycoprotein expressed on the surface of multiple immune cell types (*Ferrero and Malavasi, 2002*), and a target of therapies for multiple myeloma (*Dima et al., 2020*; *Sanchez et al., 2016*). In both cases, peak-to-protein correlations identified many CRMs were missed with peak-to-gene correlation alone. The converse is also observed in our experiment: reliably detected genes may enable detection of some regulatory regions that are not identified using protein detection alone – and peak-to-gene links based on scRNA-seq can be generated genomewide for all expressed genes, whether or not they are targeted by the epitope panel.

## Comparisons of TEA-seq to CITE-seq and scATAC-seq

To perform a direct assessment of TEA-seq data quality relative to previously published assays, we performed a comparison to standalone 10x scATAC-seq on permeabilized cells, CITE-seq using the 10x 3′ scRNA-seq kit, and both unstained nuclei and unstained permeabilized cells on the 10x Multiome kit (RNA + ATAC). All samples originated from the same pool of thawed and FACS sorted PBMCs (with dead cells, debris, and neutrophils removed). Separate aliquots were taken for scATAC-seq and the two unstained Multiome kit preps, while CITE-seq and TEA-seq samples were taken from the same pool of ADT-stained PBMCs (*Figure 4—figure supplement 4a*). In each case, 15,000 cells were used as input, and we downsampled a single well from the resulting libraries to equal sequencing depth for a fair comparison ($200 \times 10^6$ total fragments per well for RNA and ATAC libraries; $48 \times 10^6$ for ADT libraries). We performed four wells of TEA-seq to generate a large reference dataset (shown in *Figure 4*), and one well for each other assay. For all comparisons presented in *Figure 4—figure supplement 4*, we used a single well of TEA-seq. In these direct comparisons, we found that TEA-seq results appeared qualitatively similar to CITE-seq and scATAC-seq across assays (*Figure 4—figure supplement 4b, e, i*). The alignment method used for these datasets allowed us to quantify the number of RNA UMIs that aligned to intronic vs. exonic regions of the transcriptome. We found that the fraction of reads mapping to exons was lower than that found in whole-cell CITE-seq data, similar to that seen using a lysed-cell nuclei preparation (median fraction of reads with exonic mapping: TEA-seq, 51.8%, Nuclei Multiome, 52.2%; CITE-seq, 69.5% *Figure 4—figure supplement 4c*), suggesting the primary source of RNAs in TEA-seq is intron-containing nuclear transcripts. We assessed gene detection by calculating the fraction of cells with UMIs > 0 for each gene and plotted comparisons between TEA-seq and 10x Multiome nuclei (*Figure 4—figure supplement 4d*) as well as CITE-seq (*Figure 4—figure supplement 4g*). We found that gene detection was most like nuclei and lower in many cases than CITE-seq. One group of genes stands out as

particularly low, originating from ribosomal gene transcripts (with gene symbols starting with RPL and RPS, highlighted in blue). In comparisons of scATAC-seq metrics, we found that TEA-seq performed comparably to standalone permeabilized cell scATAC-seq or Multiome RNA + ATAC using permeabilized cells, with higher FRIP than purified nuclei used for the same assay (*Figure 4—figure supplement 4e, f*). Finally, in comparisons of protein detection by ADTs, we found that antibody performance in TEA-seq varied by target. Many bimodal distributions were retained (e.g., CD4, CD8a, CD86), while some separations were less apparent (e.g., CD45RO and CD45RA; *Figure 4— figure supplement 4h*), resulting in reduced separation in UMAP analysis for some types (i.e., naïve vs. mature CD4 cells), though most major cell types were able to be separated (*Figure 4—figure supplement 4i*). Together, these results show that TEA-seq data is of high quality but may not always be the method of choice if a particular modality is of greatest interest to researchers. The strength of TEA-seq is its ability to measure all three aspects – transcription, epitopes, and accessibility – simultaneously on each cell, enabling deeper insights into the connections between regulation, expression, and function as demonstrated in *Figure 4*.

## Discussion

Multimodal data collection from permeabilized PBMCs will be of use to many researchers in the immunology field and beyond who seek to get the most high-quality data from precious clinical samples and to facilitate cross-datastream integration with flow cytometry. We found that isotonic cell permeabilization allows the generation of scATAC-seq libraries with high quality as measured by FRIP and low nucleosomal content, suggesting that chromatin state in the nucleus is unperturbed (*Figure 1*, *Figure 2*). While a previous study utilized FACS followed by scATAC on individually sorted cells (Pi-ATAC; *Chen et al., 2018c*), the use of permeabilized cells enables simultaneous interrogation of chromatin accessibility state in the nucleus and the functional state of cells based on their cell surface proteins (ICICLE-seq, *Figure 3*) together with mRNA (TEA-seq, *Figure 4*) at unprecedented scale for the first time. Our methods utilize permeabilized cells to allow the addition of protein measurements that provide a direct link between high-quality scRNA-seq data and scATAC-seq data using truly paired methods extending beyond SNARE-seq (*Chen et al., 2019*) and SHARE-seq (*Ma et al., 2020*) by taking advantage of 10x Genomics Single Cell Multiome sequencing reagents as a platform for nucleic acid capture. The use of a droplet-based method allows multimodal measurements at a much higher scale by capturing thousands of single cells at a time. We expect that the datasets generated by our study will be useful beyond researchers studying PBMC biology, including those seeking to develop new analytical tools to remove background noise from existing sn/scATAC-seq data (e.g., SCALE, *Xiong et al., 2019*, or scOpen, *Li et al., 2019b*) and tools to accurately identify links across modalities (e.g., AI-TAC, *Tasaki et al., 2020*).

We have focused on PBMCs during this initial development and demonstration of the utility of TEA-seq. PBMCs are a widely used source of clinically relevant heterogeneous cell populations for monitoring clinical disease status and research. As such, TEA-seq optimized for PBMCs will provide unprecedented biological insight into the molecular underpinnings of human immune health and disease. It is likely that the use of TEA-seq in other cell types will require sample-specific optimization. For example, the requirement for neutrophil depletion may be limited to studies related to the immune system or immune-infiltrated tissues. However, the value of debris and dead cell removal is likely to be applicable to disaggregated tissue samples from biopsies (*Donlin et al., 2018*). Likewise, researchers should take care when assessing fragment length distributions as a quality control metric. In PBMCs, we found that high abundance of mononucleosomal fragments may indicate neutrophil-driven background, but this indicator of poor data quality may not be observed in other cell types or tissues. Early versions of nuclei-based ATAC-seq suffered issues high levels of mitochondrial reads (*Corces et al., 2017*), which may be used to profile clonal populations based on their mitochondrial genomes (*Lareau et al., 2020*). Here, we have used low concentrations of digitonin to selectively permeabilize cholesterol-containing membranes, thereby leaving the inner mitochondrial membrane intact (*Adam et al., 1990*; *Colbeau et al., 1971*). Different cell types have been shown to have differing sensitivity to digitonin concentrations, perhaps reflecting natural variance in membrane cholesterol composition (*Holden and Horton, 2009*). As such, cell permeabilization for TEA-seq may need to be optimized for other cell types. This can be readily tested using bulk ATAC-seq

and by measuring library complexity, FRIP score, and mitochondrial contamination to determine data quality.

In addition to the Methods presented in this article, we are maintaining an online protocol to aid the research community in further development and adoption of TEA-seq on the Protocols.io platform (https://doi.org/10.17504/protocols.io.bqagmsbw). Additional development could allow simultaneous measurement of additional aspects of cell biology (e.g., CpG sequencing methods, scCUT&Tag; *Bartosovic et al., 2020*), as well as improvements to the modalities in the TEA-seq trio (e.g., s3-ATAC, *Mulqueen et al., 2021*). In direct comparisons of TEA-seq to existing methods (CITE-seq and scATAC-seq), we found that the sensitivity of gene detection was lower and distributions of antibody detections were altered relative to CITE-seq; however, ATAC-seq data quality remained comparable to the standalone assay (*Figure 4—figure supplement 4*). We anticipate that simultaneous, truly multimodal measurement across molecular compartments will be an essential tool to expand our view of the full picture of immune cell state in health and disease, with many applications extending throughout the genomics research community.

# Materials and methods

## Key resources table

| Reagent type (species) or resource | Designation | Source or reference | Identifiers | Additional information |
|---|---|---|---|---|
| Biological sample (*Homo sapiens*) | Primary peripheral blood mononuclear cells | BioIVT, Westbury, NY, USA | SERATRIALS-18002; sample BRH1291132 | WIRB protocol # 20190318; Ficoll-purified |
| Biological sample (*Homo sapiens*) | Primary peripheral blood mononuclear cells | BioIVT, Westbury, NY, USA | SERATRIALS-18002; sample HMN85396 | WIRB protocol # 20190318; leukapheresis-purified |
| Biological sample (*Homo sapiens*) | Primary peripheral blood mononuclear cells | BioIVT, Westbury, NY, USA | SERATRIALS-18002; sample RG1131 | WIRB protocol # 20190318; Ficoll-purified |
| Biological sample (*Homo sapiens*) | Primary peripheral nlood mononuclear cells | Bloodworks NW, Seattle, WA, USA | BT001; sample 5716BW | WIRB protocol # 20141589; Ficoll-purified |
| Biological sample (*Homo sapiens*) | Primary peripheral blood mononuclear cells | Bloodworks NW, Seattle, WA, USA | BT001; sample 7811BW | WIRB protocol # 20141589; Ficoll-purified |
| Antibody | Biotin anti-human CD15 (SSEA-1) antibody (mouse monoclonal) | BioLegend | Cat# 301913, RRID:AB_2561325 | (0.15 µL per million cells) |
| Antibody | BUV395 anti-human CD14 antibody (mouse monoclonal) | BD Biosciences | Cat# 563561, RRID:AB_2744288 | (1 µL per million cells) |
| Antibody | BUV496 mouse anti-human CD45 antibody (mouse monoclonal) | BD Biosciences | Cat# 750179, RRID:AB_2868405 | (1 µL per million cells) |
| Antibody | BUV563 mouse anti-human CD15 antibody (mouse monoclonal) | BD Biosciences | Cat# 741417, RRID:AB_2868406 | (1 µL per million cells) |
| Antibody | BUV615 mouse anti-human CD56 antibody (mouse monoclonal) | BD Biosciences | Cat# 613001, RRID:AB_2868413 | (2 µL per million cells) |
| Antibody | Brilliant Violet 650 anti-human CD3 antibody (mouse monoclonal) | BioLegend | Cat# 300468, RRID:AB_2629574 | (2 µL per million cells) |
| Antibody | Mouse anti-human CD19 High Parameter Custom Antibody BB790-P Conjugate (mouse monoclonal) | BD Biosciences | Cat# 624296 | Custom reagent; conjugate of mouse anti-human CD19 (clone HIB19) and BB790-P (1 µL per million cells) |

*Continued on next page*

*Continued*

| Reagent type (species) or resource | Designation | Source or reference | Identifiers | Additional information |
|---|---|---|---|---|
| Antibody | PE/Cy5 anti-human CD16 antibody (mouse monoclonal) | BioLegend | Cat# 302009, RRID:AB_314209 | (1 µL per million cells) |
| Antibody | BUV395 mouse anti-human CD3 antibody (mouse monoclonal) | BD Biosciences | Cat# 563546, RRID:AB_2744387 | Custom reagent; conjugate of mouse anti-human CD3 (clone UCHT1) and BUV395 (1:50 in 100 µL volume) |
| Antibody | Mouse anti-human CD45RA High Parameter Custom Antibody BUV615-P Conjugate (mouse monoclonal) | BD Biosciences | Cat# 624297 | Custom reagent; conjugate of mouse anti-human CD45RA (clone HI100) and BUV615-P (1:5 in 100 µL volume) |
| Antibody | BUV661 mouse anti-human CD14 antibody (mouse monoclonal) | BD Biosciences | Cat# 741684, RRID:AB_2868407 | (1:50 in 100 µL volume) |
| Antibody | BUV737 mouse anti-human CD8 antibody (mouse monoclonal) | BD Biosciences | Cat# 749367, RRID:AB_2868408 | (1:5 in 100 µL volume) |
| Antibody | BUV805 mouse anti-human CD11c antibody (mouse monoclonal) | BD Biosciences | Cat# 742005, RRID:AB_2868409 | (1:33 in 100 µL volume) |
| Antibody | BV421 mouse anti-human CD25 antibody (mouse monoclonal) | BD Biosciences | Cat# 562442, RRID:AB_11154578 | (1:50 in 100 µL volume) |
| Antibody | BV480 mouse anti-human CD4 (mouse monoclonal) | BD Biosciences | Cat# 566104, RRID:AB_2739506 | (1:5 in 100 µL volume) |
| Antibody | BV605 mouse anti-Human CD16 (FcγRIII) antibody (mouse monoclonal) | BD Biosciences | Cat# 563172, RRID:AB_2744297 | (1:50 in 100 µL volume) |
| Antibody | Brilliant Violet 650 anti-human CD123 antibody (mouse monoclonal) | BioLegend | Cat# 306020, RRID:AB_2563827 | (1:50 in 100 µL volume) |
| Antibody | Brilliant Violet 711 anti-human CD127 (IL-7Rα) antibody (mouse monoclonal) | BioLegend | Cat# 351328, RRID:AB_2562908 | (1:50 in 100 µL volume) |
| Antibody | BV750 mouse anti-human IgD antibody (mouse monoclonal) | BD Biosciences | Cat# 747484, RRID:AB_2868411 | (1:20 in 100 µL volume) |
| Antibody | BV786 mouse anti-human neuropilin-1 (CD304) antibody (mouse monoclonal) | BD Biosciences | Cat# 743132, RRID:AB_2741299 | (1:50 in 100 µL volume) |
| Antibody | BB515 mouse anti-human CD141 antibody (mouse monoclonal) | BD Biosciences | Cat# 565084, RRID:AB_2739058 | (1:33 in 100 µL volume) |
| Antibody | CD11b monoclonal antibody, PerCP-Cy5.5 conjugated (rat monoclonal) | BD Biosciences | Cat# 561114, RRID:AB_2033995 | (1:5 in 100 µL volume) |
| Antibody | PE anti-human CD27 antibody (mouse monoclonal) | BioLegend | Cat# 302808, RRID:AB_314300 | (1:20 in 100 µL volume) |
| Antibody | PE/Dazzle 594 anti-human TCRα/β antibody (mouse monoclonal) | BioLegend | Cat# 306726, RRID:AB_2566599 | (1:33 in 100 µL volume) |

*Continued*

| Reagent type (species) or resource | Designation | Source or reference | Identifiers | Additional information |
|---|---|---|---|---|
| Antibody | Mouse anti-CD34 monoclonal antibody, PE-Cy5 conjugated (mouse monoclonal) | BD Biosciences | Cat# 555823, RRID:AB_396152 | (1:20 in 100 μL volume) |
| Antibody | PE/Cy7 anti-human CD197 (CCR7) antibody (mouse monoclonal) | BioLegend | Cat# 353226, RRID:AB_11126145 | (1:33 in 100 μL volume) |
| Antibody | APC anti-human CD38 antibody (mouse monoclonal) | BioLegend | Cat# 356606, RRID:AB_2561902 | (1:20 in 100 μL volume) |
| Antibody | APC-R700 mouse anti-human CD56 (NCAM-1) antibody (mouse monoclonal) | BD Biosciences | Cat# 565139, RRID:AB_2744429 | (1:20 in 100 μL volume) |
| Antibody | APC/Cyanine7 anti-human HLA-DR antibody (mouse monoclonal) | BioLegend | Cat# 307618, RRID:AB_493586 | (1:20 in 100 μL volume) |
| Antibody | TotalSeq-A0062 anti-human CD10 antibody (mouse monoclonal) | BioLegend | Cat# 312231, RRID:AB_2734286 | (0.5 μg per million cells) |
| Antibody | TotalSeq-A0161 anti-human CD11b antibody (mouse monoclonal) | BioLegend | Cat# 301353, RRID:AB_2734249 | (0.05 μg per million cells) |
| Antibody | TotalSeq-A0053 anti-human CD11c antibody (mouse monoclonal) | BioLegend | Cat# 371519, RRID:AB_2749971 | (0.025 μg per million cells) |
| Antibody | TotalSeq-A0064 anti-human CD123 antibody (mouse monoclonal) | BioLegend | Cat# 306037, RRID:AB_2749977 | (0.1 μg per million cells) |
| Antibody | TotalSeq-A0390 anti-human CD127 (IL-7Rα) antibody (mouse monoclonal) | BioLegend | Cat# 351352, RRID:AB_2734366 | (0.025 μg per million cells) |
| Antibody | TotalSeq-A0081 anti-human CD14 antibody (mouse monoclonal) | BioLegend | Cat# 301855, RRID:AB_2734254 | (0.2 μg per million cells) |
| Antibody | TotalSeq-A0163 anti-human CD141 (thrombomodulin) antibody (mouse monoclonal) | BioLegend | Cat# 344121, RRID:AB_2783229 | (0.1 μg per million cells) |
| Antibody | TotalSeq-A0083 anti-human CD16 antibody (mouse monoclonal) | BioLegend | Cat# 302061, RRID:AB_2734255 | (0.05 μg per million cells) |
| Antibody | TotalSeq-A0408 anti-human CD172a (SIRPα) antibody (mouse monoclonal) | BioLegend | Cat# 372109, RRID:AB_2783285 | (0.25 μg per million cells) |
| Antibody | TotalSeq-A0144 anti-human CD185 (CXCR5) antibody (mouse monoclonal) | BioLegend | Cat# 356937, RRID:AB_2750356 | (0.125 μg per million cells) |
| Antibody | TotalSeq-A0050 anti-human CD19 antibody (mouse monoclonal) | BioLegend | Cat# 302259, RRID:AB_2734256 | (0.2 μg per million cells) |

*Continued on next page*

*Continued*

| Reagent type (species) or resource | Designation | Source or reference | Identifiers | Additional information |
|---|---|---|---|---|
| Antibody | TotalSeq-A0242 anti-human CD192 (CCR2) antibody (mouse monoclonal) | BioLegend | Cat# 357229, RRID:AB_2750501 | (0.5 µg per million cells) |
| Antibody | TotalSeq-A0148 anti-human CD197 (CCR7) antibody (mouse monoclonal) | BioLegend | Cat# 353247, RRID:AB_2750357 | (0.5 µg per million cells) |
| Antibody | TotalSeq-A0181 anti-human CD21 antibody (mouse monoclonal) | BioLegend | Cat# 354915, RRID:AB_2750006 | (0.05 µg per million cells) |
| Antibody | TotalSeq-A0180 anti-human CD24 antibody (mouse monoclonal) | BioLegend | Cat# 311137, RRID:AB_2750374 | (0.5 µg per million cells) |
| Antibody | TotalSeq-A0085 anti-human CD25 antibody (mouse monoclonal) | BioLegend | Cat# 302643, RRID:AB_2734258 | (0.08 µg per million cells) |
| Antibody | TotalSeq-A0056 anti-human CD269 (BCMA) antibody (mouse monoclonal) | BioLegend | Cat# 357521, RRID:AB_2749974 | (0.5 µg per million cells) |
| Antibody | TotalSeq-A0191 anti-mouse/rat/human CD27 antibody (Armenian hamster monoclonal) | BioLegend | Cat# 124235, RRID:AB_2750344 | (0.125 µg per million cells) |
| Antibody | TotalSeq-A0171 anti-human/mouse/rat CD278 (ICOS) antibody (Armenian hamster monoclonal) | BioLegend | Cat# 313555, RRID:AB_2800824 | (0.01 µg per million cells) |
| Antibody | TotalSeq-A0088 anti-human CD279 (PD-1) antibody (mouse monoclonal) | BioLegend | Cat# 329955, RRID:AB_2734322 | (0.1 µg per million cells) |
| Antibody | TotalSeq-A0034 anti-human CD3 antibody (mouse monoclonal) | BioLegend | Cat# 300475, RRID:AB_2734246 | (0.05 µg per million cells) |
| Antibody | TotalSeq-A0406 anti-human CD304 (neuropilin-1) antibody | BioLegend | Cat# 354525, RRID:AB_2783261 | (0.1 µg per million cells) |
| Antibody | TotalSeq-A0830 anti-human CD319 (CRACC) antibody (mouse monoclonal) | BioLegend | Cat# 331821, RRID:AB_2800872 | (0.5 µg per million cells) |
| Antibody | TotalSeq-A0410 anti-human CD38 antibody (mouse monoclonal) | BioLegend | Cat# 356635, RRID:AB_2800967 | (0.05 µg per million cells) |
| Antibody | TotalSeq-A0176 anti-human CD39 antibody (mouse monoclonal) | BioLegend | Cat# 328233, RRID:AB_2750005 | (0.05 µg per million cells) |
| Antibody | TotalSeq-A0072 anti-human CD4 antibody (mouse monoclonal) | BioLegend | Cat# 300563, RRID:AB_2734247 | (0.1 µg per million cells) |
| Antibody | TotalSeq-A0031 anti-human CD40 antibody (mouse monoclonal) | BioLegend | Cat# 334346, RRID:AB_2749968 | (0.25 µg per million cells) |

*Continued on next page*

*Continued*

| Reagent type (species) or resource | Designation | Source or reference | Identifiers | Additional information |
|---|---|---|---|---|
| Antibody | TotalSeq-A0063 anti-human CD45RA antibody (mouse monoclonal) | BioLegend | Cat# 304157, RRID:AB_2734267 | (0.0625 µg per million cells) |
| Antibody | TotalSeq-A0087 anti-human CD45RO antibody (mouse monoclonal) | BioLegend | Cat# 304255, RRID:AB_2734268 | (0.1 µg per million cells) |
| Antibody | TotalSeq-A0047 anti-human CD56 (NCAM) antibody (mouse monoclonal) | BioLegend | Cat# 362557, RRID:AB_2749970 | (0.15 µg per million cells) |
| Antibody | TotalSeq-A0166 anti-human CD66b antibody (mouse monoclonal) | BioLegend | Cat# 392905, RRID:AB_2750372 | (0.25 µg per million cells) |
| Antibody | TotalSeq-A0394 anti-human CD71 antibody (mouse monoclonal) | BioLegend | Cat# 334123, RRID:AB_2800884 | (0.025 µg per million cells) |
| Antibody | TotalSeq-A0005 anti-human CD80 antibody (mouse monoclonal) | BioLegend | Cat# 305239, RRID:AB_2749958 | (0.5 µg per million cells) |
| Antibody | TotalSeq-A0006 anti-human CD86 antibody (mouse monoclonal) | BioLegend | Cat# 305443, RRID:AB_2734273 | (0.05 µg per million cells) |
| Antibody | TotalSeq-A0080 anti-human CD8a antibody (mouse monoclonal) | BioLegend | Cat# 301067, RRID:AB_2734248 | (0.2 µg per million cells) |
| Antibody | TotalSeq-A0156 anti-human CD95 (Fas) antibody (mouse monoclonal) | BioLegend | Cat# 305649, RRID:AB_2750368 | (0.1 µg per million cells) |
| Antibody | TotalSeq-A0352 anti-human FcεRIα antibody (mouse monoclonal) | BioLegend | Cat# 334641, RRID:AB_2750503 | (0.5 µg per million cells) |
| Antibody | TotalSeq-A0159 anti-human HLA-DR antibody (mouse monoclonal) | BioLegend | Cat# 307659, RRID:AB_2750001 | (0.25 µg per million cells) |
| Antibody | TotalSeq-A0384 anti-human IgD antibody (mouse monoclonal) | BioLegend | Cat# 348243, RRID:AB_2783238 | (0.05 µg per million cells) |
| Antibody | TotalSeq-A0090 Mouse IgG1, κ isotype Ctrl antibody (mouse monoclonal) | BioLegend | Cat# 400199, RRID:AB_2868412 | (0.5 µg per million cells) |
| Antibody | TotalSeq-A0136 anti-human IgM antibody (mouse monoclonal) | BioLegend | Cat# 314541, RRID:AB_2749992 | (0.05 µg per million cells) |
| Antibody | TotalSeq-A0153 anti-human KLRG1 (MAFA) antibody (mouse monoclonal) | BioLegend | Cat# 367721, RRID:AB_2750373 | (0.25 µg per million cells) |
| Antibody | TotalSeq-A0584 anti-human TCR Vα24-Jα18 (iNKT cell) antibody (mouse monoclonal) | BioLegend | Cat# 342923, RRID:AB_2783227 | (0.5 µg per million cells) |

*Continued on next page*

*Continued*

| Reagent type (species) or resource | Designation | Source or reference | Identifiers | Additional information |
|---|---|---|---|---|
| Antibody | TotalSeq-A0581 anti-human TCR Vα7.2 antibody (mouse monoclonal) | BioLegend | Cat# 351733, RRID:AB_2783246 | (0.05 µg per million cells) |
| Antibody | TotalSeq-A0224 anti-human TCR α/β antibody (mouse monoclonal) | BioLegend | Cat# 306737, RRID:AB_2783167 | (0.125 µg per million cells) |
| Antibody | TotalSeq-A0139 anti-human TCR γ/δ antibody (mouse monoclonal) | BioLegend | Cat# 331229, RRID:AB_2734325 | (0.5 µg per million cells) |
| Sequence-based reagent | MOSAIC_Bot | Integrated DNA Technologies | Tn5 complex oligo | 5'-/5phos/CTGTCTCTTATACACATCT-'3; standard desalting |
| Sequence-based reagent | Tn5ME-s7_Top | Integrated DNA Technologies | Tn5 complex oligo | 5'-GTCTCGTGGGCTCGGAGATGTGTATAAGAGACAG-3'; standard desalting |
| Sequence-based reagent | Poly-A Top-L | Integrated DNA Technologies | Tn5 complex oligo | 5'-TTTTTTTTTTTTTTTTTTTTTTTTTTTTTTVNAGATGTGTATAAGAGACAG -3'; standard desalting |
| Sequence-based reagent | SI-P5-22 | Integrated DNA Technologies | PCR primer | 5'-AATGATACGGCGACCACCGAGATCTACACTCTTTCCCTACACGACGCTCTTCCGATCT-3'; standard desalting |
| Sequence-based reagent | F BC primer | Integrated DNA Technologies | PCR primer | 5'-CTACACGACGCTCTTCCGATCT-3'; standard desalting |
| Sequence-based reagent | ADT-Rev-AMP | Integrated DNA Technologies | PCR primer | 5'-CCTTGGCACCCGAGAATTCC-3'; standard desalting |
| Sequence-based reagent | TruSeq R1 Seq primer | Integrated DNA Technologies | PCR primer | 5'-ACACTCTTTCCCTACACGACGCTCTTCCGATCT-3'; standard desalting |
| Sequence-based reagent | ADT-i7 primer | Integrated DNA Technologies | PCR primer | 5'-CAAGCAGAAGACGGCATACGAGATNNNNNNNNGTGACTGGAGTTCCTTGGCACCCGAGAATTCC*A-3'; PAGE purification; * = phosphorothioate bond; N's indicate barcode position |
| Sequence-based reagent | SI-PCR-Oligo | Integrated DNA Technologies | PCR primer | 5'-AATGATACGGCGACCACCGAGATCTACACTCTTTCCCTACACGACGCTC-3'; standard desalting |
| Peptide, recombinant protein | Tn5 transposase | Beta Lifescience | TN5-BL01 | |
| Commercial assay or kit | Chromium Next GEM Single Cell 3' GEM, Library & Gel Bead Kit v3.1 | 10x Genomics | 1000121 | |
| Commercial assay or kit | Next GEM Chip G Single Cell Kit | 10x Genomics | 1000120 | |
| Commercial assay or kit | Single Index Kit T Set A | 10x Genomics | 1000213 | |
| Commercial assay or kit | Chromium Next GEM Single Cell ATAC Library & Gel Bead Kit v1.1 | 10x Genomics | 1000175 | |
| Commercial kit or assay | Next GEM Chip H Single Cell Kit | 10x Genomics | 1000161 | |

*Continued on next page*

*Continued*

| Reagent type (species) or resource | Designation | Source or reference | Identifiers | Additional information |
|---|---|---|---|---|
| Commercial kit or assay | Single Index Kit N Set A | 10x Genomics | 1000212 | |
| Commercial kit or assay | Chromium Next GEM Single Cell Multiome ATAC + Gene Expression Reagent Bundle | 10x Genomics | 1000283 | |
| Commercial kit or assay | Next GEM Chip J Single Kit | 10x Genomics | 1000234 | |
| Commercial kit or assay | Dual Index Kit TT Set A | 10x Genomics | 1000215 | |
| Commercial kit or assay | Quant-iT PicoGreen dsDNA Assay Kit | Thermo Fisher Scientific | P7589 | |
| Commercial kit or assay | Library Quantification Kit, Illumina Platform | KAPA Biosystems | KK4844 | |
| Commercial kit or assay | Bioanalyzer high-sensitivity DNA analysis | Agilent Technologies | 5067-4626 | |
| Software, algorithm | R | The R Foundation | RRID:SCR_001905 | v3.6.3 and >4.0.2 |
| Software, algorithm | ArchR | Jeffrey Granja and Ryan Corces | | v1.0.1 |
| Software, algorithm | BarCounter | Elliott Swanson; this study | | |
| Software, algorithm | cellranger | 10x Genomics | | v5.0.0 |
| Software, algorithm | cellranger-atac | 10x Genomics | | v1.1.0 and v1.2.0 |
| Software, algorithm | cellranger-arc | 10x Genomics | | v1.0.0 |
| Software, algorithm | Seurat | Paul Hoffman, Satija Lab, and Collaborators | RRID:SCR_016341 | v4.0-beta |
| Software, algorithm | FlowJo | BD | | v10.7 |
| Other | Digitonin | MP Biomedicals | 0215948082 | |
| Other | Human TruStain FcX | BioLegend | 422302 | |
| Other | ViaStain acridine orange/propidium iodide solution | Nexcelom Bioscience | CS2-0106-25mL | |
| Other | Dynabeads MyOne SILANE | Thermo Fisher Scientific | 37002D | |
| Other | Fixable Viability Stain 510 | BD Biosciences | 564406 | |
| Other | IGEPAL-CA630 | Sigma | I8896 | CAS 9002-93-1 |
| Other | SPRIselect | Beckman Coulter | B23319 | |
| Other | NEBNext Ultra II Q5 Master Mix | New England Biolabs | M0544 | |
| Other | KAPA HiFi HotStart ReadyMix | KAPA Biosystems | KM2602 | |
| Other | Protector RNase Inhibitor | Sigma-Aldrich | 03335399001 | |
| Other | Fixable Viability Stain 510 | BD Biosciences | Cat# 564406, RRID:AB_2869572 | |

## Sample collection and preparation
### Sample collection and processing
Biological specimens were purchased from BioIVT as cryopreserved PBMCs and Bloodworks NW as freshly drawn whole blood. All sample collections were conducted by BioIVT and Bloodworks NW

under IRB-approved protocols, and all donors signed informed consent forms. See *Supplementary file 7* for a list of sources and samples used for data displayed in each figure.

PBMCs sourced from BioIVT were isolated using either Ficoll-Paque or leukapheresis. Following isolation, PBMCs were subjected to RBC lysis, washing, and counting. PBMC aliquots were cryopreserved in Cryostor CS10 (StemCell Technologies, 07930) and stored in vapor phase liquid nitrogen.

For fresh blood samples from Bloodworks NW, PBMC processing occurred in-house. Blood tubes were pooled, gently swirled until fully mixed, about 30 times, and diluted with an equivalent volume of room temperature PBS (Thermo Fisher Scientific, 14190235). PBMCs were isolated using one or more Leucosep tubes (Greiner Bio-One, 227290) loaded with 15 mL of Ficoll Premium (GE Healthcare, 17-5442-03) to which a 3 mL cushion of PBS had been slowly added on top of the Leucosep barrier. Diluted whole blood (24–30 mL) was slowly added to each tube and spun at $1000 \times g$ for 10 min at 20°C with no brake (Beckman Coulter Avanti J-15RIVD with JS4.750 swinging bucket, B99516). PBMCs were recovered from the Leucosep tube by quickly pouring all volume above the barrier into a sterile 50 mL conical tube (Corning, 352098). A 15 mL cold phosphate-buffered saline (PBS) +0.2% bovine serum albumin (BSA) (Sigma, A9576; 'PBS+BSA') was added and the cells were pelleted at $400 \times g$ for 5–10 min at 4–10°C. The supernatant was quickly decanted, the pellet dispersed by flicking the tube, and the cells washed with 25–50 mL cold PBS+BSA. Cell pellets were combined as needed, the cells were pelleted as before, supernatant quickly decanted, and residual volume was carefully aspirated. PBMCs were resuspended in 1 mL cold PBS+BSA per 15 mL whole blood processed and counted with a ViCell (Beckman Coulter) using VersaLyse reagent (Beckman Coulter, A09777) or with a Cellometer Spectrum Cell Counter (Nexcelom) using ViaStain acridine orange/propidium iodide solution (Nexcelom, C52-0106-5). PBMCs were cryopreserved in Cryostor10 (StemCell Technologies, 07930) or 90% fetal bovine serum (FBS, Thermo Fisher Scientific, 10438026)/10% dimethyl sulfoxide (DMSO, Fisher Scientific, D12345) at $5 \times 10^6$ cells/mL by slow freezing in a Coolcell LX (VWR, 75779-720) overnight in a −80°C freezer followed by transfer to liquid nitrogen.

## Cell thawing

Cryopreserved PBMCs were removed from liquid nitrogen storage and thawed in a 37°C water bath for 3–5 min until no ice was visible. Cells were diluted to 10 mL in 37°C AIM V medium (Gibco, 12055091) with the first 3 mL added dropwise. Cells were then washed once with 10 mL DPBS without calcium and magnesium (Corning, 21-031 CM) supplemented with 0.2% w/v BSA (Sigma-Aldrich, A2934). Cells were counted on a Cellometer Spectrum Cell Counter (Nexcelom) using ViaStain acridine orange/propidium iodide solution (Nexcelom, C52-0106-5) and stored on ice.

## FACS neutrophil depletion

To remove dead cells, debris, and neutrophils, PBMC samples were sorted by FACS prior to nuclei isolation or cell permeabilization. Cells were incubated with Fixable Viability Stain 510 (BD, 564406) for 15 min at room temperature and washed with AIM V medium (Gibco, 12055091) plus 25 mM HEPES before incubating with TruStain FcX (BioLegend, 422302) for 5 min on ice, followed by staining with anti-CD45 (BioLegend, 304038) and anti-CD15 (BD, 562371) antibodies for 20 min on ice. Cells were washed with AIM V medium plus 25 mM HEPES and sorted on a BD FACSAria Fusion. A standard viable CD45+ cell gating scheme was employed; FSC-A vs. SSC-A (to exclude sub-cellular debris), two FSC-A doublet exclusion gates (FSC-W followed by FSC-H), dead cell exclusion gate (BV510 LIVE/DEAD negative) followed by CD45+ inclusion gate. Neutrophils (defined as SSC^high, CD15+) were then excluded in the final sort gate (*Figure 1—figure supplement 3*). An aliquot of each post-sort population was used to collect 50,000 events to assess post-sort purity.

## Magnetic bead neutrophil depletion

Bead-based neutrophil depletion was performed using a biotin conjugated monoclonal anti-CD15 antibody in combination with streptavidin-coated magnetic beads. A high neutrophil content (approximately 1.1%) Ficoll-isolated PBMC sample was processed to evaluate efficacy, and a low neutrophil leukapheresis-isolated PBMC sample was processed to control for off-target effects. Briefly, $1 \times 10^7$ PBMCs were resuspended in 100 µL of chilled DPBS without calcium and magnesium (Corning, 21-031 CM) supplemented with 0.2% w/v BSA (Sigma-Aldrich, A2934). A 10 µL TruStain

FcX (BioLegend 422302) was added to the cell suspension, mixed by pipette, and incubated on ice for 10 min. Anti-CD15 antibody (BioLegend, 301913) was added to the cell suspension, mixed by pipette, and incubated on ice for 30 min. Following antibody binding, 25 µL of Dynabeads MyOne Streptavidin T1 magnetic beads (Invitrogen, 65601) was added to the cell suspension, mixed by pipette, and incubated at room temperature for 5 min. The cell suspension was then diluted with 900 µL of room temperature Dulbecco's phosphate-buffered saline (DPBS) +0.2% w/v BSA and placed on an EasySep magnet (Stemcell Technologies, 18103) for 3 min. The supernatant (approximately 1 mL) was transferred to a new tube and stored on ice until further processing.

Non-depleted and neutrophil-depleted PBMCs from each sample were analyzed by flow cytometry using an eight-color panel to assess the effects of the bead-based depletion on major PBMC populations. For each sample and condition, $1 \times 10^6$ cells were centrifuged ($750 \times g$ for 5 min at 4℃) using a swinging bucket rotor (Beckman Coulter Avanti J-15RIVD with JS4.750 swinging bucket, B99516), the supernatant was removed using a vacuum aspirator pipette, and the cell pellet was resuspended in 100 µL of DPBS without calcium and magnesium (Corning, 21-031 CM) supplemented with 0.2% w/v BSA (Sigma-Aldrich, A2934). Cells were incubated with Fixable Viability Stain 510 (BD, 564406) and TruStain FcX (BioLegend, 422302) for 30 min on ice, and washed in chilled FACS buffer (DPBS, 0.2% w/v BSA, 0.1% sodium azide; VWR, BDH7465-2). Cells were stained with a cocktail of antibodies (*Supplementary file 1*) including 10 µL of Brilliant Stain Buffer Plus (BD, 566385) at a staining volume of 100 µL for 30 min on ice, then washed twice with chilled FACS buffer. Cells were passed through 35 µm Falcon Cell Strainers (Corning, 352235) and analyzed on a BD FACS Symphony flow cytometer. Gating analysis was performed using FlowJo cytometry software (version 10.7).

A sequential gating scheme was used to identify viable singlet CD45+/CD15+/CD16+ neutrophils: (1) a time vs. SSC-A gate (to confirm that no abnormalities occurred in the fluidics), (2) a FSC-A vs. SSC-A gate (to exclude sub-cellular debris), (3) two FSC-A doublet exclusion gates (FSC-W followed by FSC-H), (3) a dead cell exclusion gate (BV510 LIVE/DEAD negative), and (4) a CD45+ inclusion gate. Neutrophils were defined as either SSC-A$^{high}$/CD15+ or CD15+/CD16+. The neutrophil population defined by SSC-A$^{high}$/CD15+ was larger than that defined by CD15+/CD16+ due to the presence of some contaminating CD15$^{low}$ monocytes. Therefore, we used the CD45+/CD15+/CD16+ gate for subsequent analysis including summary statistics (*Figure 1—figure supplement 4* and *Supplementary file 2*).

## Standard nuclei isolation

Isolation of nuclei suspensions was performed according to the Demonstrated Protocol: Nuclei Isolation for Single Cell ATAC Sequencing (10x Genomics, CG000169 Rev C). Briefly, $8 \times 10^5$ to $1 \times 10^6$ cells were added to a 1.5 mL low binding tube (Eppendorf, 022431021) and centrifuged ($300 \times g$ for 5 min at 4℃) using a swinging bucket rotor (Beckman Coulter Avanti J-15RIVD with JS4.750 swinging bucket, B99516). The supernatant was removed using a vacuum aspirator pipette, and the cell pellet was resuspended in 100 µL of chilled 10x Genomics Nuclei Isolation Buffer (10 mM Tris-HCl pH 7.4, 10 mM NaCl, 3 mM MgCl$_2$, 0.1% Tween-20, 0.1 % NP-40 Substitute CAS 9016-45-9 [BioVision 2127-50], 0.01% digitonin [MP Biomedicals 0215948082], 1% BSA) by pipette-mixing 10 times. Cells were incubated on ice for 3 min, followed by dilution with 1 mL of chilled 10x Wash Buffer (10 mM Tris-HCl pH 7.4, 10 mM NaCl, 3 mM MgCl$_2$, 0.1% Tween-20 [Bio-Rad 1610781], 1% BSA) by pipette-mixing five times. Nuclei were centrifuged ($500 \times g$ for 3 min at 4℃), and the supernatant was slowly removed using a vacuum aspirator pipette. Nuclei were resuspended in chilled 1x Nuclei Buffer (10x Genomics, 2000207) to a target concentration of 3000–6000 nuclei per µL. Nuclei suspensions were passed through 35 µm Falcon Cell Strainers (Corning, 352235) and counted on a Cellometer Spectrum Cell Counter (Nexcelom) using ViaStain acridine orange/propidium iodide staining solution (Nexcelom, C52-0106-5).

## Nuclei isolation optimization

In addition to 10x Nuclei Isolation Buffer (10xNIB), we tested an alternative Nuclei Isolation Buffer (ANIB) as described previously (*Mulqueen et al., 2019*) 10 mM Tris-HCl pH 7.4, 10 mM NaCl, 3 mM MgCl$_2$, 0.1% Tween-20, 0.1% IGEPAL CAS 9002-93-1 (Sigma, I8896), 1x Protease Inhibitor (Roche, 11836170001). For each buffer, we generated a titration series of detergent concentrations relative

to the concentrations described above but did not alter the concentration of other buffer ingredients: 1×, 0.5×, 0.25×, 0.1× for 10×NIB, and 1× and 0.1× for ANIB. The resulting nuclei were imaged using an EVOS M5000 Imaging System (Thermo Fisher Scientific, AMF5000) in transmitted light mode at ×40 magnification to visually evaluate nuclear integrity (*Figure 1—figure supplement 1*). The 1 × 10 ×NIB, 0.25 × 10 ×NIB, 0.1 × 10 ×NIB, and 1× ANIB were used for 10X scATAC-seq (*Figure 1—figure supplement 2*).

## Cell permeabilization

We prepared a 5% w/v digitonin stock by diluting powdered digitonin (MP Biomedicals, 0215948082) with 100% DMSO (Fisher Scientific, D12345) and creating 20 µL aliquots, which were stored at −20°C. To permeabilize, 1,000,000 cells were added to a 1.5 mL low binding tube (Eppendorf, 022431021) and centrifuged (400×*g* for 5 min at 4°C) using a swinging bucket rotor (Beckman Coulter Avanti J-15RIVD with JS4.750 swinging bucket, B99516). The supernatant was removed using a vacuum aspirator pipette, and the cell pellet was resuspended in 100 µL of chilled isotonic Perm Buffer (20 mM Tris-HCl pH 7.4, 150 mM NaCl, 3 mM MgCl$_2$, 0.01% digitonin) by pipette-mixing 10 times. Cells were incubated on ice for 5 min, after which they were diluted with 1 mL of isotonic Wash Buffer (20 mM Tris-HCl pH 7.4, 150 mM NaCl, 3 mM MgCl$_2$) by pipette-mixing five times. Cells were centrifuged (400×*g* for 5 min at 4°C) using a swinging bucket rotor, and the supernatant was slowly removed using a vacuum aspirator pipette. The cell pellet was resuspended in chilled TD1 buffer (Illumina, 15027866) by pipette-mixing to a target concentration of 2300–10,000 cells per µL. Cells were passed through 35 µm Falcon Cell Strainers (Corning, 352235) and counted on a Cellometer Spectrum Cell Counter (Nexcelom) using ViaStain acridine orange/propidium iodide solution (Nexcelom, C52-0106-5). For optimization, we used varying final digitonin concentrations in the Perm Buffer: 0.01% w/v, 0.05% w/v, 0.1% w/v, and 0.2% w/v. The optimal concentration observed was 0.01% w/v.

## snATAC-seq and scATAC-seq

### sn/scATAC-seq library preparation

scATAC-seq libraries were prepared according to the Chromium Single Cell ATAC v1.1 Reagent Kits User Guide (CG000209 Rev B) with several modifications. In total, 15,000 cells or nuclei were loaded into each tagmentation reaction. Nuclei were brought up to a volume of 5 µL in 1x Nuclei Buffer (10x Genomics, 2000207), mixed with 10 µL of a transposition master mix consisting of ATAC Buffer B (10x Genomics, 2000193) and ATAC Enzyme (Tn5 transposase; 10x Genomics, 2000123). Permeabilized cells were brought up to a volume of 9 µL in TD1 buffer (Illumina, 15027866) and mixed with 6 µL of Illumina TDE1 Tn5 transposase (Illumina, 15027916). Transposition was performed by incubating the prepared reactions on a C1000 Touch thermal cycler with 96-Deep Well Reaction Module (Bio-Rad, 1851197) at 37°C for 60 min, followed by a brief hold at 4°C. A Chromium NextGEM Chip H (10x Genomics, 2000180) was placed in a Chromium Next GEM Secondary Holder (10x Genomics, 3000332), and 50% glycerol (Teknova, G1798) was dispensed into all unused wells. A master mix composed of Barcoding Reagent B (10x Genomics, 2000194), Reducing Agent B (10x Genomics, 2000087), and Barcoding Enzyme (10x Genomics, 2000125) was then added to each sample well, pipette-mixed, and loaded into row 1 of the chip. Chromium Single Cell ATAC Gel Beads v1.1 (10x Genomics, 2000210) were vortexed for 30 s and loaded into row 2 of the chip, along with Partitioning Oil (10x Genomics, 2000190) in row 3. A 10x Gasket (10x Genomics, 370017) was placed over the chip and attached to the Secondary Holder. The chip was loaded into a Chromium Single Cell Controller instrument (10x Genomics, 120270) for GEM generation. At the completion of the run, GEMs were collected and linear amplification was performed on a C1000 Touch thermal cycler with 96-Deep Well Reaction Module: 72°C for 5 min, 98°C for 30 s, 12 cycles of 98°C for 10 s, 59°C for 30 s, and 72°C for 1 min.

GEMs were separated into a biphasic mixture through addition of Recovery Agent (10x Genomics, 220016), and the aqueous phase was retained and removed of barcoding reagents using Dynabeads MyOne SILANE (10x Genomics, 2000048) and SPRIselect reagent (Beckman Coulter, B23318) bead clean-ups. Sequencing libraries were constructed by amplifying the barcoded ATAC fragments in a sample indexing PCR consisting of SI-PCR Primer B (10x Genomics, 2000128), Amp Mix (10x Genomics, 2000047), and Chromium i7 Sample Index Plate N, Set A (10x Genomics, 3000262) as

described in the 10x scATAC User Guide. Amplification was performed in a C1000 Touch thermal cycler with 96-Deep Well Reaction Module: 98°C for 45 s, for 9–11 cycles of 98°C for 20 s, 67°C for 30 s, 72°C for 20 s, with a final extension of 72°C for 1 min. Final libraries were prepared using a dual-sided SPRIselect size-selection cleanup. SPRIselect beads were mixed with completed PCR reactions at a ratio of 0.4× bead:sample and incubated at room temperature to bind large DNA fragments. Reactions were incubated on a magnet, and the supernatant was transferred and mixed with additional SPRIselect reagent to a final ratio of 1.2× bead:sample (ratio includes first SPRI addition) and incubated at room temperature to bind ATAC fragments. Reactions were incubated on a magnet, the supernatant containing unbound PCR primers and reagents was discarded, and DNA-bound SPRI beads were washed twice with 80% v/v ethanol. SPRI beads were resuspended in Buffer EB (Qiagen, 1014609), incubated on a magnet, and the supernatant was transferred, resulting in final, sequencing-ready libraries.

## sn/scATAC-seq sequencing

Final libraries were quantified using a Quant-iT PicoGreen dsDNA Assay Kit (Thermo Fisher Scientific, P7589) on a SpectraMax iD3 (Molecular Devices). Library quality and average fragment size were assessed using a Bioanalyzer (Agilent, G2939A) High Sensitivity DNA chip (Agilent, 5067-4626). Libraries were sequenced on the Illumina NovaSeq platform with the following read lengths: 51 nt read 1, 8 nt i7 index, 16 nt i5 index, and 51 nt read 2.

## sn/scATAC-seq data processing

Demultiplexing of raw base call files into FASTQ files was performed using 10x cellranger-atac mkfastq (10x Genomics v.1.1.0). To assess samples at an equal sequencing depth, FASTQ files were downsampled to a uniform total raw read count among compared samples: $2 \times 10^8$ fragments for comparison of nuclei and cells across FACS conditions (*Figures 1* and *2*); $1.25 \times 10^8$ fragments for optimization experiments (*Figure 1—figure supplement 2*) due to lower available total read depth after sequencing. A 10x cellranger-atac count was used to process sequencing reads by performing adapter trimming and sequence alignment to the GRCh38 (hg38) reference genome (refdata-cellranger-atac-GRCh38-1.1.0). The output files fragments.tsv.gz and singlecell.csv were utilized for downstream processing and quality control analysis.

To evaluate quality control metrics across all scATAC-seq datasets, we utilized bedtools (v2.29.1) and GNU parallel (*Tange, 2011*) v20161222 to generate overlap counts and feature count matrices for a panel of reference genomic regions: 518,766 peaks from a previous study of PBMCs by scATAC-seq (*Lareau et al., 2019*; supplementary file GSE123577_pbmc_peaks.bed.gz from GEO accession GSE123577) were converted from hg19 to hg38 coordinates using the UCSC liftOver tool (*Hinrichs et al., 2006*; kent source v402) and used to compute a standardized fraction of reads in peaks score for cells in each dataset (FRIP); 33,496 transcription start site regions (TSS ±2 kb) from Hg38 ENSEMBL release 93 (*Yates et al., 2020*) were filtered to select genes used in the 10x Genomics cellranger GRCh38 reference for scRNA-seq (refdata-cellranger-GRCh38-3.0.0) and used to compute the fraction of reads in TSS (FRITSS); and a set of 3,591,898 reference DNase hypersensitive sites from ENCODE (*Meuleman et al., 2020*; ENCODE file ID ENCFF503GCK) were used to assess distal regulatory element accessibility. In addition, we generated tiled window counts across the genome in 5k, 20k, 100k bins.

## sn/scATAC-seq quality control

Custom R scripts were used to assess and filter preprocessed scATAC-seq data along a variety of quality metrics. Cells or nuclei with >1,000 uniquely aligned fragments, FRIP >0.2, FRITSS > 0.2, and fraction of fragments overlapping ENCODE reference regions > 0.5 were retained for downstream analysis. Cells or nuclei that passed these QC cutoffs were used to generate sparse count matrices and filtered fragments.tsv.gz files for downstream analysis.

To examine aggregate TSS accessibility, we selected fragments from fragments.tsv.gz that overlapped TSS regions describe above (TSS ±2 kb). For plotting, fragments were separated using cell barcodes (and fragment length in the case of *Figure 1—figure supplement 5*) into separate groups. Fragment positions were converted to positions relative to TSS (sensitive to transcript strand orientation), and the number of fragments overlapping each position was calculated.

To examine CTCF motif accessibility, CTCF motif locations were obtained from genome-wide motif scans of non-redundant TF motifs (*Vierstra, 2020*; https://resources.altius.org/jvierstra/projects/motif-clustering/releases/v1.0). Motifs were filtered to select CTCF motifs that overlapped ENCODE reference regions (*Meuleman et al., 2020*; ENCODE file ID ENCFF503GCK). Selected were ranked by their MOODS match, and the top 100,000 motifs were selected for analysis. Motif locations were expanded to a total of 4 kb centered on the middle of each CTCF motif (using the resize function from GenomicRanges in R). Fragments from cells that passed QC filtering were converted to target site duplication (TSD) center positions (+5 bp from the 5′ end and −4 bp from the 3′ end of each fragment). All TSD centers that overlapped expanded CTCF motif regions were selected, and the number of TSD centers that overlapped each position relative to the CTCF motif was calculated (sensitive to CTCF motif strand orientation).

## sn/scATAC-seq saturation analysis

To assess sequencing library saturation, we used the fragments.tsv.gz output file from cellranger-atac, which provides the observed count of each unique fragment position, to calculate how many fragments were observed with each frequency. This count-of-counts table was used as input for the R package preseqR function preseqR.rSAC.bootstrap, which predicts the number of unique species (unique ATAC fragments in our case) that are represented at least r times based on an initial sample (*Daley and Smith, 2013*; *Deng et al., 2018*). For each dataset, we ran 100 rounds of bootstrapped estimations at 10 million-fragment steps, from 0 to 2 billion total fragments per sample. To assess the fraction of additional unique fragments obtained at each 10 million read step (as shown in *Figure 1—figure supplement 3*), we subtracted the estimated number of unique fragments from the number of unique fragments at the previous step and divided by 10 million (the number of additional raw sequenced fragments).

## sn/scATAC-seq dimensionality reduction

For 2D projections of scATAC-seq data, we used binarized sparse matrices of 20 kb window accessibility across the hg38 genome (excluding mitochondrial regions, chrM). Independently for each dataset, we selected features found in >3% of cells/nuclei, weighted features using term frequency-inverse document frequency, log-transformed the resulting weights, and performed principle component analysis (PCA) using singular value decomposition to generate 50 reduced dimensions as described previously (*Cusanovich et al., 2018*; *Hill, 2019*). We then removed the first PC, which was strongly correlated with the number of available fragments and retained the remaining PCs up to PC 30. For display, we further reduced the dimensionality of selected PCs using UMAP (*Becht et al., 2019*; *McInnes et al., 2018*) (R package uwot, v0.1.8, parameters: scale = TRUE, min_dist = 0.2).

## sn/scATAC-seq cell-type labeling

Labeling of scATAC-seq datasets was performed using the ArchR package (*Granja et al., 2020*) v0.9.4. In brief, filtered fragments.tsv.gz files after quality control were used to generate an ArchR GeneScore matrix and a tiled genome feature matrix for each dataset. Cells were grouped by performing iterative latent semantic indexing (LSI) on the tile matrix, followed by the shared nearest-neighbor clustering approach implemented in Seurat (*Stuart et al., 2019*) v3.1.5. GeneScore data was then used to compare scATAC-seq clusters to a labeled reference scRNA-seq dataset consisting of 9380 PBMCs generated by 10x Genomics, with labels provided by the Satija lab (https://www.dropbox.com/s/zn6khirjafoyyxl/pbmc_10k_v3.rds?dl=0) using ArchR's implementation of the FindTransferAnchors method from Seurat. The best-scoring labels for each scATAC-seq cluster were used for downstream analysis and display (*Figure 2*), and label transfer scores for individual cells were used to compare label transfer between methods (*Figure 2c*).

## sn/scATAC-seq peak analysis

After labeling cell types in each dataset, peaks for each cell type were generated using the ArchR functions addGroupCoverages and addReproduciblePeakSet. Within each dataset, peaks scores from each pair of cell types were compared using getMarkerFeatures performed in each direction separately by swapping foreground and background cell types. To identify differentially accessible markers, the Wilcoxon rank sum test was used (testMethod = 'wilcoxon'), and multiple hypothesis

test correction was performed using the Benjamini and Hochburg method for false discovery rate estimation (FDR; *Benjamini and Hochberg, 1995*). Differentially accessible peaks (DAPs) from each comparison were selected with filter string "FDR <= 0.05 and Log2FC >= log2(1.5)".

To identify enriched TFBS motifs, CIS-BP motif annotations (*Weirauch et al., 2014*) were attached to each peak identified by ArchR using the addMotifAnnotations. Marker peaks for each cell type were identified using getMarkerFeatures without specifying foreground and background groups using the Wilcoxon rank sum test and FDR correction, as described above. Enriched motifs were identified using the ArchR function peakAnnoEnrichment, which performs a hypergeometric test for enrichment of motif annotations in marker peaks, with FDR correction for multiple hypothesis testing (parameters: peakAnnotation = 'Motif', cutOff = "FDR <= 0.01 and Log2FC >= log2(1.5)"). Up to the top 10 enriched motifs for each cell type were plotted using the ArchR function plotEnrichHeatmap.

## Cell-type flow cytometry

To assess cell-type proportions, PBMCs were analyzed with a 25-color immunophenotyping flow cytometry panel. $1 \times 10^6$ thawed PBMCs were centrifuged ($750 \times g$ for 5 min at 4°C) using a swinging bucket rotor (Beckman Coulter Avanti J-15RIVD with JS4.750 swinging bucket, B99516), the supernatant was removed using a vacuum aspirator pipette, and the cell pellet was resuspended in DPBS without calcium and magnesium (Corning 21-031 CM). Cells were incubated with Fixable Viability Stain 510 (BD, 564406) and TruStain FcX (BioLegend, 422302) for 30 min at 4°C, then washed with chilled PBS+0.2% BSA (Sigma, A9576; 'PBS+BSA'). Cells were stained with a cocktail of antibodies (*Supplementary file 3*) at a staining volume of 100 μL for 30 min at 4°C, then washed with PBS +0.2% BSA. Fixation was performed by resuspending cells in 100 μL of 4% paraformaldehyde (Electron Microscopy Sciences, 15713) and incubating for 15 min at 25°C, protected from light. Following fixation, cells were washed twice with PBS+0.2% BSA and resuspended in 100 μL PBS (without BSA). Stained cells were analyzed on a five-laser Cytek Aurora spectral flow cytometer. Spectral unmixing was calculated with pre-recorded reference controls using Cytek SpectroFlo software (version 2.0.2). Cell types were quantified by traditional bivariate gating analysis performed with FlowJo cytometry software (version 10.7, *Figure 2—figure supplement 1*).

## ICICLE-seq

### Oligonucleotide sources

Custom oligonucleotides for Tn5 complexes, indexing, and amplification were ordered from Integrated DNA Technologies. Sequences and modifications are provided in *Supplementary file 5*. ADT i7 indexing primers contain a phosphorothioate bond linking the final two 3′ bases and were purified using polyacrylamide gel electrophoresis (PAGE). The MOSAIC_Bot oligo used in Tn5 complexing contains a 5′ phosphate group and was purified using standard desalting. All remaining oligos were produced without chemical modifications and purified using standard desalting.

### Tn5 complexing

The assembly of Tn5 transposomes was performed as previously described (*Mulqueen et al., 2019*). DNA complexes containing mosaic-end sequences with either a poly-T or Nextera R2N 5′ overhang (Poly-T Top-L/MOSAIC_Bot, Tn5ME-s7_Top/MOSAIC_Bot) were created by annealing equimolar amounts of top and bottom oligos (*Figure 3—figure supplement 1* and *Supplementary file 5*) on a C1000 Touch thermal cycler with 96-Deep Well Reaction Module (Bio-Rad, 1851197) at 95°C for 5 min followed by 5°C decreases every 2 min until the temperature reached 20°C. Oligos were annealed at a concentration of 16 μM in 2× Dialysis Buffer (100 mM HEPES-KOH pH 7.5 [Teknova, 550000-016], 200 mM NaCl, 0.2 mM EDTA, 2 mM DTT [IBI Scientific, 21040], 0.2% Triton X-100 [Sigma-Aldrich, T8787], 20% glycerol [Teknova, G1798]). Annealed complexes were mixed 1:1 for a final concentration of 8 nM. Tn5 transposase (Beta Lifescience, TN5-BL01) was supplemented with 5 M NaCl at a final volume ratio of 1:8 NaCl to Tn5. The resulting NaCl/Tn5 mixture was mixed with the annealed complexes at a volume ratio of 1.2:1 ratio of DNA complexes to Tn5 and incubated at 25°C for 60 min to form final, reaction-ready Tn5 complexes, which were stored at −20°C until use.

## Antibody staining

PBMCs were depleted of neutrophils, dead cells, and debris through FACS as described above. $2 \times 10^6$ sorted PBMCs were centrifuged ($400 \times g$ for 5 min at 4°C) using a swinging bucket rotor (Beckman Coulter Avanti J-15RIVD with JS4.750 swinging bucket, B99516), the supernatant was removed using a vacuum aspirator pipette, and the cell pellet was resuspended in 100 µL of DPBS without calcium and magnesium (Corning 21-031 CM) supplemented with 2% w/v BSA (Sigma-Aldrich A2934). A 10 µL TruStain FcX (BioLegend, 422302) was added and cells were incubated on ice for 10 min. A panel of 46 barcoded oligo-conjugated antibodies (BioLegend TotalSeq-A) including a mouse IgG1K isotype-negative control (*Supplementary file 6*) was added and incubated on ice for 30 min. Cells were washed three times in 4 mL of DPBS plus 2% BSA to remove unbound antibodies and used as input into cell permeabilization with 0.01% digitonin as described above.

## ICICLE-seq library preparation

Transposition was performed by aliquoting 20,000 permeabilized cells in TD1 buffer (Illumina, 15027866), bringing the volume up to 9 µL in TD1 buffer, and mixing with 6 µL of Poly-T overhang Tn5 complexes. Reactions were incubated on a C1000 Touch thermal cycler with 96-Deep Well Reaction Module (Bio-Rad, 1851197) at 37°C for 120 min, followed by a brief hold at 4°C. Cell barcodes were then added to ATAC and ADTs via GEM generation using 10x Genomics 3′ RNA beads and subsequent amplification. Briefly, a Chromium Next GEM Chip G (10x Genomics, 2000177) was placed in a Chromium Next GEM Secondary Holder (10x Genomics, 3000332) and 50% glycerol (Teknova, G1798) was dispensed into all unused wells. A barcoding master mix was prepared, which consisted of NEBNext Ultra II Q5 Master Mix (New England Biolabs, M0544), Reducing Agent B (10x Genomics, 2000087), F BC Primer (0.2 µM, *Supplementary file 5*), and ADT-Rev-AMP (0.2 µM, *Supplementary file 5*). The master mix was added to each sample well, pipette-mixed, and loaded into row 1 of the chip. Chromium Single Cell 3′ v3.1 Gel Beads (10x Genomics, 2000164) were vortexed for 30 s and loaded into row 2 of the chip, along with Partitioning Oil (10x Genomics, 2000190) in row 3. A 10x Gasket (10x Genomics, 370017) was placed over the chip and attached to the Secondary Holder. The chip was loaded into a Chromium Single Cell Controller instrument (10x Genomics, 120270) for GEM generation. At the completion of the run, GEMs were collected and amplification was performed on a C1000 Touch thermal cycler with 96-Deep Well Reaction Module: 72°C for 5 min, 98°C for 30 s, 12 cycles of 98°C for 10 s, 42°C for 30 s, and 65°C for 30 s, followed by a final extension of 65°C for 1 min.

GEMs were separated into a biphasic mixture through addition of Recovery Agent (10x Genomics, 220016), the aqueous phase was retained and removed of barcoding reagents using Dynabeads MyOne SILANE (10x Genomics, 2000048) beads. Next, a dual-sided 0.6×/2.0× bead:sample SPRIselect reagent (Beckman Coulter, B23318) size-selection clean-up was performed to remove large DNA fragments and unused primers. Libraries were split into two reactions in a 3:1 ATAC:ADT ratio and amplified separately using different indexed P7 primers. ATAC fragments were amplified in a 100 µL reaction consisting of Buffer EB (Qiagen, 1014609), Amp Mix (10x Genomics, 2000047), SI-P5-22 primer (20 µM, *Supplementary file 5*), and Chromium i7 Multiplex Kit N Set A (10x Genomics, 3000262). ATAC PCR was performed in a C1000 Touch thermal cycler with 96-Deep Well Reaction Module: 98°C for 45 s, seven cycles of 98°C for 20 s, 54°C for 30 s, and 72°C for 20 s, followed by a final extension of 72°C for 1 min. ADT fragments were amplified in a 100 µL reaction consisting of Buffer EB (Qiagen, 1014609), KAPA HiFi HotStart ReadyMix (KAPA Biosystems, KM2602), SI-P5-22 primer (10 µM, *Supplementary file 5*), and ADT i7 primer (10 µM, *Supplementary file 5*). ADT PCR was performed in a C1000 Touch thermal cycler with 96-Deep Well Reaction Module: 95°C for 3 min, 15 cycles of 95°C for 20 s, 60°C for 30 s, and 72°C for 20 s, followed by a final extension of 72°C for 5 min. SPRIselect reagent cleanups were performed with a 1.2× bead:sample ratio for ADT libraries and a dual-sided size selection of 0.4×/1.2× bead:sample ratio for ATAC libraries.

## ICICLE-seq sequencing

Final libraries were quantified using qPCR (KAPA Biosystems Library Quantification Kit for Illumina, KK4844) on a CFX96 Touch Real-Time PCR Detection System (Bio-Rad, 1855195). Library quality and average fragment size were assessed using a Bioanalyzer (Agilent, G2939A) High Sensitivity DNA chip (Agilent, 5067-4626). Libraries were sequenced on the Illumina NovaSeq platform with the

following read lengths: 28 bp read 1 (Cell barcode and UMI), 8 bp i7 index, 100 bp read 2 (ATAC-seq sequence or ADT barcode). A Truseq read 1 primer (0.3 µM, *Supplementary file 5*) was included as a Custom Read 1 primer to mitigate the risk of off-target priming of the standard Illumina Nextera read 1 primer on the partial Nextera R1N sequence included in the mosaic end portion of the Poly-T Tn5 insertion.

## ICICLE-seq data preprocessing

Demultiplexing of raw base call files into FASTQ files was performed using bcl2fastq2 (Illumina v2.20.0.422). Read 2 was trimmed of adapter sequences, low-quality bases and reads, and polyA tailing using fastp (*Chen et al., 2018a*) v0.21.0 (parameters: `–adapter_sequence`=CTGTCTCTTA TACACATCT `–cut_tail –trim_poly_x`) and the resulting read 2 sequences were aligned to the GRCh38 (hg38) reference genome (Illumina iGenomes, https://support.illumina.com/sequencing/sequencing_software/igenome.html) using Bowtie 2 (*Langmead and Salzberg, 2012*; v2.3.0, parameters: –local, –sensitive, –no-unal, –phred33). Aligned reads in SAM format were filtered by alignment score (≥30) then tagged with cell barcode and UMI sequence and quality scores using custom python code (python3 v3.7.3). Barcode sequences were compared against the 10x Genomics v3 3′ GEX barcode whitelist (3M-february-2018.txt.gz). Sequences not included in the whitelist were corrected to a valid whitelist barcode by allowing a single base mismatch (Hamming distance of 1). Sequences with more than one possible match were corrected at the position with the lowest sequencing quality score. Reads with barcodes that could not be corrected were excluded from further analysis. Filtered and tagged SAM files were converted to sorted, indexed BAM files using GATK (*McKenna et al., 2010*; Broad Institute v4.1.4.0). Genomic coordinates were converted to BED format using bedtools (*Quinlan and Hall, 2010*; v2.26.0). Custom python code was used to collapse aligned fragments into a list of fragments with unique cell barcode and genomic coordinate combinations. These fragments were then written as a fragments.tsv.gz file in the format: chr, start position, end position, cell barcode, UMI count, and strand (+/–).

## ADT data preprocessing

Methods for ADT counting were developed in-house and were implemented as an optimized, highly efficient C program named BarCounter (available at https://github.com/AllenInstitute/Barcounter-release). BarCounter was used for single-cell ADT counting as follows: firstly, barcode sequences were compared against the 10x Genomics v3 3′ GEX barcode whitelist (3M-february-2018.txt.gz). Sequences not included in the whitelist were corrected to a valid whitelist barcode by allowing a single base mismatch (hamming distance of 1) at a low-quality basecall (sequencing quality score <20). Reads with barcodes that could not be corrected were excluded from further analysis. Next, ADT barcode sequences were compared against a CSV taglist containing ADT barcode/antibody associations. Antibody barcodes in the current TotalSeq-A catalog (BioLegend) have a Hamming distance from all other barcodes of at least 3. Therefore, a single base mismatch (Hamming distance of 1) was allowed. Reads containing ADT sequences that could not be assigned to an antibody in the taglist were excluded. Finally, UMI sequences that were unique within their assigned ADT for their assigned cell barcode were counted. Final ADT UMI counts were written by cell barcode to a CSV file for use in downstream analysis.

ADT features were filtered by comparison to the mouse IgG1κ isotype control, which should not bind to human cell surface proteins. The distribution of counts for each antibody was compared to the control using a Mann–Whitney test (R function wilcox.test with parameter alternative = 'greater'). Any features for which the test returned a p-value>$1\times10^{-9}$ were considered similar to the control and were removed from downstream analysis (*Supplementary file 6*).

## ICICLE-seq analysis

Aligned ICICLE-seq chromatin accessibility fragments.tsv.gz files were preprocessed as for 10× scATAC-seq samples, above. QC filtering was performed as described, with modified cutoffs: >500 uniquely aligned reads, FRIP >0.65, FRITSS > 0.2, and fraction of fragments overlapping ENCODE reference regions > 0.5 were retained for downstream analysis. For 2D projections of the scATAC-seq data, we used binarized sparse matrices of 20 kb window accessibility across the hg38 genome, selected features found in >0.5% of cells, weighted features using LogTF-IDF, and performed PCA

as described above. We then removed the first PC and retained the remaining PCs up to PC 20. UMAP was performed with adjusted parameters (scale = FALSE, min_dist = 0.2). To assign cell-type labels, filtered fragments.tsv.gz files were used as input to ArchR. ArchR functions addIterativeLSI and addGeneIntegrationMatrix (parameters transferParams = list(dims = 1:10, k.weight = 20) and nGenes = 4000) were used to transfer labels from the scRNA-seq PBMC reference described above (scATAC Cell Type Labeling).

Count matrices for ADT data were scaled for each cell by dividing by the thousands of total ADT UMIs per cell, then transformed using $Log_{10}$(scaled count + 1). Normalized features were used for PCA using the R function prcomp with default parameters. Filtered and normalized features were used as direct input to UMAP (R package uwot with parameter min_dist = 0.2) with the first two PCs from PCA used as initial coordinates to aid reproducibility of UMAP projection. Cells were clustered using a Jaccard–Louvain method (parameters k = 15, radius = 1) using UMAP coordinates. Clusters with high signal from the mouse IgG1κ isotype control antibody were removed from subsequent analysis (one cluster, n = 32 cells). The remaining clusters were manually labeled by examination of cell-type marker enrichment. The R package scratch.vis (*Tasic et al., 2018*; https://github.com/alle-ninstitute/scrattch.vis; *Graybuck, 2021a*; copy archived at swh:1:rev:aa9094fb3d92f05264-c9aa0911eb4c4967862609) was used to generate the cluster median heatmap plot and a river/alluvial plot comparing the cell-type labels obtained from ATAC-seq and ADT-based analyses.

To compare peaks between cell types, all filtered fragments for cells in each cell type were aggregated and used as input to the MACS2 peak caller (*Zhang et al., 2008*; parameters -f BED, -g hs, –no-model). The top 2500 peaks from each cell type were selected for comparison (except for Plasmablasts, for which all 592 peaks were used). A master set of peaks across all types was constructed by combining all narrowPeak results files and combining the outer coordinates of overlapping peaks (GenomicRanges function reduce). A binary matrix of peak overlaps for each cell type was generated and used to construct the peak comparison figure inspired by UpSet plots (*Lex et al., 2014*).

## TEA-seq

### TEA-seq library preparation

Consistent with ICICLE-seq, cryopreserved PBMCs were thawed, stained for FACS, depleted of neutrophils, dead cells, and debris through FACS, stained with TotalSeq-A antibodies (BioLegend, *Supplementary file 6*), and permeabilized using 0.01% digitonin as described above. ATAC and Gene Expression libraries were prepared according to the Chromium Next GEM Single Cell Multiome ATAC + Gene Expression User Guide (CG000338 Rev A) with several modifications. To generate a high-quality reference dataset, we generated four 10x Genomics wells of TEA-seq. For each well, transposition was performed by diluting 15,400 permeabilized cells to a final volume of 5 μL in isotonic Tagmentation Buffer (20 mM Tris-HCl pH 7.4, 150 mM NaCl, 3 mM $MgCl_2$, RNase Inhibitor 1 U/μL; Lucigen NxGen, F83923-1). Cells were mixed with 10 μL of a transposition master mix consisting of 7 μL ATAC Buffer B (10x Genomics, 2000193) and 3 μL ATAC Enzyme B (Tn5 transposase; 10x Genomics, 2000265) per reaction. Transposition was performed by incubating the prepared reactions on a C1000 Touch thermal cycler with 96–Deep Well Reaction Module (Bio-Rad, 1851197) at 37°C for 60 min, followed by a brief hold at 4°C. A Chromium NextGEM Chip J (10x Genomics, 2000264) was placed in a Chromium Next GEM Secondary Holder (10x Genomics, 3000332) and 50% glycerol (Teknova, G1798) was dispensed into all unused wells. A master mix composed of Barcoding Reagent Mix (10x Genomics, 2000267), Reducing Agent B (10x Genomics, 2000087), Template Switch Oligo (10x Genomics, 3000228), and Barcoding Enzyme Mix (10x Genomics, 2000266) was then added to each sample well, pipette-mixed, and loaded into row 1 of the chip. Chromium Single Cell Multiome Gel Beads v1.1 (10x Genomics, 2000261) were vortexed for 30 s and loaded into row 2 of the chip, along with Partitioning Oil (10x Genomics, 2000190) in row 3. A 10x Gasket (10x Genomics, 3000072) was placed over the chip and attached to the Secondary Holder. The chip was loaded into a Chromium Single Cell Controller instrument (10x Genomics, 120270) for GEM generation. At the completion of the run, GEMs were collected, and reverse transcription and barcoding were performed by incubating GEMs on a C1000 Touch thermal cycler with 96–Deep Well Reaction Module at 37°C for 45 min, 25°C for 30 min, followed by a brief hold at 4°C. Upon completion of the GEM incubation, 5 μL of Quenching Agent (10x Genomics, 2000269) was immediately added and mixed with the reaction solution.

GEMs were separated into a biphasic mixture through addition of Recovery Agent (10x Genomics, 220016), the aqueous phase was retained and cleared of barcoding reagents using Dynabeads MyOne SILANE (10x Genomics, 2000048) and 2.0× bead:sample ratio SPRIselect reagent (Beckman Coulter, B23318) cleanups, incubating for 10 min after each bead addition step.

Barcoded ATAC and cDNA fragments were amplified using a PCR reaction consisting of 45 µL of template, 50 µL of Pre-Amp Mix (10x Genomics, 2000270), 5 µL of Pre-Amp Primers (10x Genomics, 2000271), and 1 µL of ADT-Rev-AMP (0.2 µM, *Supplementary file 5*). Amplification was performed in a C1000 Touch thermal cycler with 96–Deep Well Reaction Module: 72℃ for 5 min, 98℃ for 3 min, for seven cycles of 98℃ for 20 s, 63℃ for 30 s, and 72℃ for 1 min, with a final extension of 72℃ for 1 min. Amplified fragments were purified using a 2.0× bead:sample ratio SPRIselect reagent (Beckman Coulter, B23318) bead clean-up, incubating for 10 min after bead addition.

Sequencing ready ATAC libraries were constructed by amplifying barcoded ATAC fragments in a sample indexing PCR consisting of SI-PCR Primer B (10x Genomics, 2000128), Amp Mix (10x Genomics, 2000047), and Sample Index Plate N, Set A (10x Genomics, 3000427) as described in the 10x Multiome User Guide. Amplification was performed in a C1000 Touch thermal cycler with 96–Deep Well Reaction Module: 98℃ for 45 s, then nine cycles of 98℃ for 20 s, 67℃ for 30 s, and 72℃ for 20 s, with a final extension of 72℃ for 1 min. Final ATAC libraries were prepared using a dual-sided 0.6×/1.6× bead:sample SPRIselect reagent (Beckman Coulter, B23318) size-selection clean-up.

cDNA fragments were amplified using a PCR reaction consisting of 35 µL of template, 50 µL of Amp Mix (10x Genomics, 2000047), 15 µL of cDNA Primers (10x Genomics, 2000089), and 1 µL of ADT-Rev-AMP (2 µM, *Supplementary file 5*). Amplification was performed in a C1000 Touch thermal cycler with 96–Deep Well Reaction Module: 98℃ for 3 min, then eight cycles of 98℃ for 15 s, 63℃ for 20 s, and 72℃ for 1 min, with a final extension of 72℃ for 1 min. cDNA and ADT fragments were separated using a dual-sided SPRIselect reagent (Beckman Coulter, B23318) size-selection clean-up. Large cDNA fragments were retained in an initial 0.6× bead:sample SPRIselect incubation, reactions were incubated on a magnet, and small unbound fragments containing ADTs were transferred to new wells where they were subjected to an additional 2.0× bead:sample SPRIselect cleanup.

Consistent with ICICLE-seq, ADT fragments were amplified in a 15 cycle indexing PCR, substituting SI-PCR-Oligo primer (10 µM, *Supplementary file 5*) for SI-P5-22. Final ADT libraries were prepared using a dual-sided 1.6× bead:sample SPRIselect reagent (Beckman Coulter, B23318) clean-up.

cDNA Fragmentation, End-Repair, and A-tailing were performed in a joint reaction containing 25% of cDNA sample as described in the 10x Multiome User Guide. Briefly, cDNA was diluted in Buffer EB (Qiagen, 1014609) and combined with master mix of Fragmentation Buffer (10x Genomics, 2000091) and Fragmentation Enzyme (10x Genomics, 2000090). The reaction was incubated in a C1000 Touch thermal cycler with 96–Deep Well Reaction Module: 4℃ start, 32℃ for 5 min, and 65℃ for 30 min, followed by a brief hold at 4℃. Reactions were purified using a dual-sided 0.6×/0.8× bead:sample SPRIselect reagent (Beckman Coulter, B23318) size-selection clean-up. Ligation was performed by mixing each sample with a master mix consisting of Ligation Buffer (10x Genomics, 2000092), DNA Ligase (10x Genomics, 220110), and Adapter Oligos (10x Genomics, 2000094), and incubating in a C1000 Touch thermal cycler with 96–Deep Well Reaction Module for 15 min at 20℃, followed by a brief hold at 4℃. Ligation reactions were purified using a 0.8× bead:sample SPRIselect reagent (Beckman Coulter, B23318) clean-up.

Sequencing ready Gene Expression libraries were constructed by amplifying cDNA fragments in a sample indexing PCR consisting of Amp Mix (10x Genomics, 2000047) and Dual Index TT Set A (10x Genomics, 3000431) as described in the 10x Multiome User Guide. Amplification was performed in a C1000 Touch thermal cycler with 96–Deep Well Reaction Module: 98℃ for 45 s, for 14 cycles of 98℃ for 20 s, 54℃ for 30 s, and 72℃ for 20 s, with a final extension of 72℃ for 1 min. Final Gene Expression libraries were prepared using a dual-sided 0.6×/0.8× bead:sample SPRIselect reagent (Beckman Coulter, B23318) size-selection clean-up.

A bench protocol for TEA-seq is available on protocols.io at https://www.protocols.io/view/tea-seq-bqagmsbw.

## TEA-seq sequencing

Final libraries were quantified using qPCR (KAPA Biosystems Library Quantification Kit for Illumina, KK4844) on a CFX96 Touch Real-Time PCR Detection System (Bio-Rad, 1855195). Library quality and average fragment size were assessed using a Bioanalyzer (Agilent, G2939A) High Sensitivity DNA chip (Agilent, 5067-4626). Libraries were sequenced on the Illumina NovaSeq platform with the following read lengths: 50 bp read 1, 10 bp i7 index, 16 bp i5 index, 90 bp read 2.

## TEA-seq data processing

Demultiplexing of raw base call files into FASTQ files was performed using bcl2fastq2 (Illumina v2.20.0.422, parameters: `–create-fastq-for-index-reads`, `–minimum-trimmed-read-length=8`, `–mask-short-adapter-reads=8`, `–ignore-missing-positions`, `–ignore-missing-filter`, `–ignore-missing-bcls`, `-r 24 w 24 p 80`). The `–use-bases-mask` option was used to adjust the read lengths for each library type as follows: Y28n*,I10,I10n*,Y90n* for Gene Expression, Y50n*,I8n*,Y16,Y50n* for ATAC, and Y28n*,I8n*,n*,Y90n* for ADT.

## TEA-seq analysis

For analysis of TEA-seq data alone, the full available sequencing depth from each library for each well was used for alignment. RNA and ATAC libraries were aligned using cellranger-arc software (v1.0.0, 10x Genomics) against 10x genomics reference refdata-cellranger-arc-GRCh38-2020-A using default parameters. ADT barcode counting was performed using BarCounter as described above for ICICLE-seq. After alignment, ATAC-seq data was used as input for scATAC-seq Quality Control analysis and filtering, as described above. The 10x droplet barcodes that were called cells by the cellranger-arc software passed ATAC-seq QC criteria (>2,500 unique fragments, FRIP >0.2, FRITSS > 0.2, and fraction of fragments overlapping ENCODE reference regions > 0.5) and had >500 genes detected were used for downstream analysis.

Selected scATAC-seq barcodes were further processed using ArchR (v1.0.1) for Iterative LSI, Clustering, Group Coverage computation, Reproducible Peak Set annotation, and sparse Peak Matrix generation using default settings according to the ArchR documentation (https://archrproject.com). We then used the Peak Matrix as input for an additional round of Iterative LSI based on these features (Peak LSI), then used the ArchR addUMAP function for two-dimensional UMAP projection.

Next, the scRNA-seq count matrix generated by cellranger-arc, the Peak Matrixand, Peak LSI data (for analysis), and UMAP projection, and Gene Score Matrix (for visualization) generated using ArchR and the ADT count matrix were filtered and combined as assays in a Seurat object (Seurat v4.0.0-beta), and dimensionality reduction for scRNA-seq and ADTs was carried out using parameters specific to each assay as described below.

For scRNA-seq, the matrix was normalized (NormalizeData function, default settings), variable features were identified (FindVariableFeatures function, defaults), the data was scaled (ScaleData function, defaults), PCA was performed (RunPCA function, defaults), and a UMAP projection was generated (RunUMAP, dims = 1:30). For label transfer, we performed SCTransform to normalize the data in order to match the Seurat Multimodal Reference Dataset for PBMCs (available from the Satija lab at https://atlas.fredhutch.org/data/nygc/multimodal/pbmc_multimodal.h5seurat, *Hao et al., 2020*). Label transfer was then performed using the Seurat functions FindTransferAnchors and TransferData as described in the Seurat v4 Vignettes. For consistency with other figures in the paper, we mapped the reference labels in the 'celltype.l2' categories onto the cell-type categories presented in the reference used for scATAC-seq label transfer, above.

For ADT counts, the data was normalized (NormalizeData function, scale.factor = 1000) and scaled (ScaleData function, defaults), then used directly as inputs for UMAP projection (RunUMAP function).

For scATAC-seq data, we imported the results from ArchR for presentation and analysis, described above.

## Multi-modal weighted nearest-neighbors analysis

In order to perform weighted nearest neighbors (WNN) analysis with input from all three assays above, we extended the functions provided by Seurat v4.0.0-beta (*Hao et al., 2020*), which utilize only two assays, to allow an arbitrary number of reduced dimension sets to be used as input. To do

so, we performed within-modality and cross-modality predictions and computed cross-modal affinities as described by Y. Hao and S. Hao et al. We then generalized the computation of cell-specific modality affinity ratio, $S$, for each cell $i$, in each modality, $\alpha$, using a small value for $\varepsilon$ ($10^{-4}$) as suggested by Y. Hao and S. Hao et al.

$$S_\alpha(i) = \frac{\theta_\alpha(x_i, \hat{x}_i, knn_\alpha)}{\max\limits_{\beta \neq \alpha}\left\{\theta_\alpha(x_i, \hat{x}_i, knn_\beta)\right\} + \varepsilon}$$

Here, $x_i$ and $x_{i,knn_\alpha}$ represent a generic form of the actual and predicted values of the low-dimensionality representation of each cell in each modality, as described for RNA and protein in Y. Hao and S. Hao et al. (e.g., $r_i$ and $r_{i,knn_\alpha}$ for scRNA), and $\beta$ is used to represent all modalities that are not the modality under consideration (e.g., ADT and scATAC-seq when $\alpha$ is RNA). The formula for $\theta_\alpha\left(x_i, x_{i,knn_\alpha}\right)$ and $\theta_\alpha\left(x_i, x_{i,knn_\beta}\right)$ is given in Y. Hao and S. Hao et al.

With these multiple $S_\alpha(i)$ values, we then compute the cell-specific modality weights, $w_\alpha(i)$ as

$$w_\alpha(i) = \frac{e^{S_\alpha(i)}}{\sum_\beta e^{S_\beta(i)}}$$

To generate the weighted pairwise cell similarities, we now sum across all modalities instead of only RNA and protein:

$$\theta_{weighted}(i,j) = \sum_\alpha w_\alpha(i)\theta_\alpha(x_i, x_j)$$

These multi-modal weighted similarities were then used to construct the WNN graph and perform WNN-based UMAP projection as described by Y. Hao and S. Hao et al. The modified version of Seurat v4.0.0-beta used for this analysis is available at https://github.com/aifimmunology/seurat (*Swanson, 2021*; copy archived at swh:1:rev:d49409e847ba787cb483d12bd5712d1083f41ac1). In the specific case of TEA-seq integration, we used scRNA-seq PCA dimensions 1:30, ATAC Peak LSI dimensions 1:30, and all normalized and scaled ADT features passing QC criteria (1:38).

## Peak linkage analysis

To identify peak-to-RNA and peak-to-protein correlations in TEA-seq data, we first generated low-dimensional embeddings based on joint analysis for RNA and ATAC data using Archr. We generated an ArchR project using all four of the TEA-seq wells generated and filtered, above. We used the 500 bp tile matrix generated by ArchR to perform LSI for ATAC data using addIterativeLSI. We then imported the RNA-seq data using import10xFeatureMatrix, added the matched RNA data to the project using addGeneExpressionMatrix, and reduced dimensions with addIterativeLSI with modified parameters (clusterParams = list(resolution = 0.2, sampleCells = 1e4, n.start = 10, varFeatures = 2500, firstSelection = 'variable', binarize = FALSE)). We then integrated LSI results using addCombinedDims, then clustered cells using the combined LSI dimensions using addClusters. We then called peaks for each cluster using addGroupCoverages and addReproduciblePeakSet, before generating a per-cell peak matrix using addPeakMatrix. Next, we recomputed the ATAC-seq LSI based on the peak matrix with addIterativeLSI and generated a refined set of joint reduced dimensions in combination with the RNA LSI results with addCombinedDims. These combined, peak-based dimensions were used to identify peak-to-RNA correlations using addPeak2GeneLinks. For loop visualizations, we filtered the peak-to-RNA correlations using getPeak2GeneLinks with adjusted parameters (corCutOff = 0.05, FDRCutOff = 0.01, resolution = 1000, returnLoops = TRUE), and results were filtered for epitope antibody targets (*Supplementary file 6*). To compute peak-to-Protein correlations, we substituted the ADT count matrix for the RNA matrix by using the ADT matrix as an input for addGeneExpressionMatrix, then re-ran the addPeak2Gene links and getPeak2GeneLinks functions with the same parameters listed, above. For comparisons shown in scatter plots, we used the base-specific (non-binned) findPeak2GeneLinks results stored in the ArchR object, and selected all peak-to-RNA or peak-to-Protein links identified for each gene with the same corCutOff and FDRCutOff parameters shown above.

## Cross-modal comparison of TEA-seq, CITE-seq, and scATAC-seq

### Joint sample processing and aliquots

To allow direct comparison of TEA-seq data to CITE-seq, scATAC-seq, and 10x Multiome RNA + ATAC performed on unstained nuclei and permeabilized cells, we thawed leukapheresis-purified PBMCs and performed dead cell, debris, and neutrophil removal by FACS as described above. We then separated this pool into separate aliquots for antibody staining (for CITE-seq and TEA-seq), nuclei isolation (for 10x Multiome RNA + ATAC), and two separate cell permeabilization aliquots (for scATAC-seq and 10x Multiome RNA + ATAC). scATAC-seq was performed using 0.01% digitonin permeabilization as described above. We then stained the CITE-seq and TEA-seq aliquot using the same 46 antibody panel as described above for ICICLE-seq and TEA-seq, and divided this stained pool for whole-cell CITE-seq (described below) and TEA-seq. Finally, we permeabilized the TEA-seq portion of this aliquot to perform TEA-seq as described above. For 10x Multiome RNA + ATAC libraries, we either permeabilized cells with 0.01% digitonin as describe above or purified nuclei as described by the 10x Genomics Nuclei Isolation for Single Cell Multiome ATAC + Gene Expression Sequencing Demonstrated Protocol (CG000365 Rev B). Protocol steps following sample preparation for TEA-seq or Multiome ATAC + Gene Expression were performed as described above. For Multiome ATAC + Gene Expression on nuclei or permeabilized cells, we excluded steps for ADT processing. A diagrammatic overview of this experimental workflow is provided in *Figure 4—figure supplement 4a*.

### CITE-seq library preparation

CITE-seq library generation was performed using 10x Genomics Single Cell 3′ v3.1 kit, based on the 10 × 3′ Gene Expression Library Construction Guide (CG000204 Rev B) and the New York Genome Center Technology Innovation Lab CITE-seq protocol (version 2019-02-13). Briefly, a Chromium Next GEM Chip G (10x Genomics, 2000177) was placed in a Chromium Next GEM Secondary Holder and 50% glycerol was dispensed into all unused wells. A barcoding master mix was prepared, which consisted of RT Reagent B (10x Genomics, 2000165), Template Switch Oligo (10x Genomics, 3000228), Reducing Agent B (10x Genomics, 2000087), and RT Enzyme C (10x Genomics, 2000085). The master mix was added to the sample well, pipette-mixed, and loaded into row 1 of the chip. Chromium Single Cell 3′ v3.1 Gel Beads (10x Genomics, 2000164) were vortexed for 30 s and loaded into row 2 of the chip, along with Partitioning Oil (10x Genomics, 2000190) in row 3. A 10x Gasket (10x Genomics, 370017) was placed over the chip and attached to the Secondary Holder. The chip was loaded into a Chromium Single Cell Controller instrument for GEM generation. GEMs were separated into a biphasic mixture through addition of Recovery Agent (10x Genomics, 220016), the aqueous phase was retained and removed of barcoding reagents using Dynabeads MyOne SILANE beads. cDNA amplification was performed using the 10x Genomics cDNA Primers (2000089) on a C1000 Touch thermal cycler with 96-Deep Well Reaction Module: 53°C for 45 min, 85°C for 5 min, followed by a hold at 4°C.

To separate ADT and cDNA libraries, we performed a 0.6× bead:sample SPRIselect reagent size selection. The bead fraction from this separation was retained as the cDNA fraction, and supernatant separation was retained as the ADT-containing fraction. To further process ADTs, we added an additional 1.4× SPRI volumes to the ADT-containing supernatant to a final ratio of 2× SPRI:sample, then retained the bead fraction of this second step. We then repeated a fresh 2× bead:sample cleanup, retaining the bead fraction as the ADT library. ADTs were amplified in a 100 µL reaction consisting of Buffer EB, KAPA HiFi HotStart ReadyMix, SI-PCR-Oligo primer (10 µM, *Supplementary file 5*), and ADT i7 primer (10 µM, *Supplementary file 5*). ADT PCR was performed in a C1000 Touch thermal cycler with 96-Deep Well Reaction Module: 95°C for 3 min, 12 cycles of 95°C for 20 s, 60°C for 30 s, and 72°C for 20 s, followed by a final extension of 72°C for 5 min. SPRIselect reagent cleanups were performed with a 1.6× bead:sample ratio to generate final ADT libraries.

cDNA Fragmentation, End-Repair, A-tailing, and indexed amplification were performed as described in the 10 × 3′ Gene Expression Library Construction Guide. These steps match the 10x Multiome guide and are described above in the TEA-seq library construction methods.

## 10x Multiome and TEA-seq library preparation

As described above, unstained FACS-sorted PBMCs were used as input to nuclei and permeabilized cell Multiome ATAC plus Gene Expression library preparation. Nuclei were isolated according to 10x Genomics Demonstrated Protocol CG000365 Rev A exactly as recommended for cryopreserved PBMCs. Cells were permeabilized as described above. Libraries were prepared according to the 10x Genomics Chromium Next GEM Single Cell Multiome ATAC plus Gene Expression user guide CG000338 Rev A.

TEA-seq libraries were prepared as described previously in the methods above.

## Sequencing of cross-modal libraries

All Multiome ATAC plus Gene Expression and TEA-seq libraries were sequenced on an Illumina NovaSeq S4 v1.0 flowcell with the following read lengths: 50 bp read 1, 10 bp i7 index, 16 bp i5 index, 90 bp read 2. Following sequencing, read lengths were adjusted to the applicable length for each datatype (ATAC, gene expression, ADT) during demultiplexing. Demultiplexing of raw base call files into FASTQ files was performed using bcl2fastq2 (Illumina v2.20.0.422, parameters: `–create-fastq-for-index-reads`, `–minimum-trimmed-read-length`=8, `–mask-short-adapter-reads`=8, `–ignore-missing-positions`, `-ignore-missing-filter`, `-ignore-missing-bcls`, -r 24 w 24 p 80). The `–use-bases-mask` option was used to adjust the read lengths for each library type as follows: Y28n*,I10,I10n*,Y90n* for Gene Expression, Y50n*,I8n*,Y16, Y50n* for ATAC, and Y28n*,I8n*,n*,Y90n* for ADT.

CITE-seq libraries were sequenced on the Illumina NextSeq platform using a high-output flowcell with the following read lengths: 28 bp read 1, 8 bp i7 index, 91 bp read 2. Demultiplexing of CITE-seq raw base call files into FASTQ files was performed using 10x Genomics cellranger mkfastq (v.3.0.2).

The scATAC library was sequenced on the Illumina NextSeq platform using a high-output flowcell with the following read lengths: 51 bp read 1, 8 bp i7 index, 16 bp i5 index, 51 bp read 2. Demultiplexing of raw base call files into FASTQ files was performed using 10x cellranger-atac mkfastq (10x Genomics v.1.1.0).

## Data analysis

To perform comparisons between these methodologies, all scRNA-seq and scATAC-seq libraries were downsampled to 200 million FASTQ entries (200 million of each sequenced read; ~20,000 reads per expected cell recovered at ~10,000 cell expected recovery), and all ADT libraries were downsampled to 48 million reads (limited by the available read count for one experiment; ~4,800reads per expected cell recovered). For scATAC-seq data, we used cellranger-atac (v 1.2.0) to align the data to 10x Genomics reference refdata-cellranger-atac-GRCh38-1.2.0. For CITE-seq data, RNA alignment was performed using cellranger (v5.0.0) with the –include-introns parameter against 10x Genomics reference refdata-gex-GRCh38-2020-A. For 10x Multiome and TEA-seq, we aligned RNA and ATAC data using cellranger-arc (v1.0.0) against 10x Genomics reference refdata-cellranger-arc-GRCh38-2020-A. To analyze CITE-seq and TEA-seq ADT libraries, we used BarCounter to compute the counts of ADT barcodes for each 10x cell barcode, as described for ICICLE-seq and TEA-seq, above.

After alignment or tabulation of each dataset, cell barcodes were filtered using modality and method-specific criteria based on inspection of the distribution of QC data for each library. Filtering was performed to remove low-quality barcodes and putative doublets. For transcription (RNA-seq data), we filtered barcodes based on number of genes detected per barcode: TEA-seq: >500 genes, <2,750 genes, 7098 barcodes retained; nuclei on 10x Multiome: >300 genes, <4,000 genes, 6717 barcodes retained; permeabilized cells on 10x Multiome: >300 genes, <2,750 genes, 8334 barcodes retained; CITE-seq: >500 genes, <5,000 genes, 7399 barcodes retained. For epitopes (ADT data), both TEA-seq and CITE-seq were filtered based on number of UMIs (>300 UMIs and <5000 UMIs): TEA-seq, 7661 barcodes retained; CITE-seq, 7493 barcodes retained. For accessibility (ATAC-seq data), barcodes were filtered using the same criteria as other scATAC-seq libraries, above (>1,000 unique fragments, FRIP > 0.2, FRITSS > 0.2, Fraction ENCODE overlap > 0.5), and doublets were removed using the ArchR package (*Granja et al., 2020*): TEA-seq: 6725 barcodes

retained; nuclei on 10x Multiome: 6541 barcodes retained; permeabilized cells on 10x Multiome: 7238 barcodes retained; standalone scATAC-seq: 11,963 barcodes retained.

After filtering, we performed modality-specific analyses to compare the performance of each method in each separate modality. For transcription, scRNA-seq datasets were analyzed using the Seurat package (v.4.0-beta, *Hao et al., 2020*; *Stuart et al., 2019*). Data was normalized using the NormalizeData function, features were selected using the FindVariableFeatures function, and data was then scaled using the ScaleData function. Scaled data for selected features was used as input for PCA to reduce dimensionality, then projected to two dimensions using the RunUMAP function on the first 30 principal component dimensions. All parameters were defaults. To label cells with PBMC cell types, we utilized the same scRNA-seq reference from the Satija lab that was used for scATAC-seq analysis, above. Label transfer was performed using FindTransferAnchors function (with the parameter reduction = 'cca'), followed by the TransferData function. Canonical correlation analysis was used instead of a PCA-based transfer because the scRNA-seq datasets were aligned using introns, while the reference was aligned using exons only. To compare gene detection across experimental methods, we computed the fraction of filtered cell barcodes with $\geq$1 UMI for each gene.

For epitope (ADT) data from filtered cell barcodes, we filtered features to remove antibodies that were similar to the Mouse IgG1κ Isotype background control. For each cell barcode, we divided the UMI counts for each feature by the number of UMI counts for the control, and flagged features with $\leq$16-fold enrichment relative to the control. We then removed the following features that were flagged in >0.5% of cells in the CITE-seq dataset: CD269 (TNFRSF17), CD304 (NRP-1), CD66b (CEACAM8), CD80, and TCR Va24-Ja18. The TEA-seq dataset was filtered to remove these features to allow comparison between TEA-seq and CITE-seq data. After feature filtering, each cell barcode was normalized by dividing the UMI counts by the total UMI count, and multiplying by 10,000. Values for each feature were then scaled using the R scale function, and all features were used as input for two-dimensional UMAP projection using the uwot R package (*Melville, 2020*).

For accessibility (scATAC-seq) data from filtered cell barcodes, we analyzed each dataset using the ArchR package (v1.0.1 *Granja et al., 2020*). For each dataset, we performed dimensionality reduction using the addIterativeLSI function (with parameter varFeatures = 50,000), then generated two-dimensional UMAP projections using the addUMAP function. To label cells, we used the addGeneIntegraitonMatrix function to compare the ArchR GeneScore matrix to the same scRNA-seq reference used for scRNA-seq, above.

## Data analysis and visualization software

Post-processing analysis of summary statistics and visualization of snATAC-seq, scATAC-seq, and ICICLE-seq was performed using R v.3.6.3 and greater (*R Development Core Team, 2020*) in the Rstudio IDE (*RStudio Team, 2020*; Integrated Development Environment for R) or using the Rstudio Server Open Source Edition as well as the following packages: for scATAC-seq-specific analyses and comparisons to scRNA-seq data, ArchR (*Granja et al., 2020*) and Seurat (*Hao et al., 2020*; *Stuart et al., 2019*); for general data analysis and manipulation, data.table (*Dowle and Srinivasan, 2019*), dplyr, Matrix (*Bates and Maechler, 2018*), matrixStats (*Bengtsson, 2018*), purrr (*Henry and Wickham, 2019*), and reshape2 (*Wickham, 2007*); for data visualization, ggplot2 (*Wickham, 2016*) and cowplot (*Wilke, 2018*); for dimensionality reduction and clustering, igraph (*Csardi and Nepusz, 2006*), RANN, and the R uwot implementation (*Melville, 2020*) of UMAP (*Becht et al., 2019*; *McInnes et al., 2018*); for manipulation of genomic region data, bedtools2 (*Quinlan and Hall, 2010*) and GenomicRanges (*Lawrence et al., 2013*); and for prediction of sequencing saturation, preseqR (*Deng et al., 2018*; *Deng et al., 2015*).

## Data and code availability

Raw scATAC-seq and ICICLE-seq, and TEA-seq data are in the process of deposition to dbGaP for controlled access. Processed data has been submitted to GEO (accession number GSE158013). Code related to processing, analysis, and visualization of data in this study is available at https://github.com/alleninstitute/aifi-swanson-teaseq (*Graybuck, 2021b*; copy archived at swh:1:rev:b88dbaf1568da1c8d6958bafd1abc8d36b214cc4).

## Acknowledgements

The authors thank Nina Kondza, Tanja Smith, Leila Shiraiwa, and Ernie Coffey for operational support, Olivia Fong for data management support, and the Allen Institute for Immunology Software Development team for computational support. The authors thank the Allen Institute founder, Paul G. Allen, for his vision, encouragement, and support.

## Additional information

### Competing interests
Cara Lord: Cara Lord is affiliated with GlaxoSmithKline. The author has no financial interests to declare. The other authors declare that no competing interests exist.

### Funding
No external funding was received for this work.

### Author contributions
Elliott Swanson, Conceptualization, Software, Formal analysis, Investigation, Methodology, Writing - original draft, Writing - review and editing; Cara Lord, Palak C Genge, Zachary Thomson, Investigation; Julian Reading, Alexander T Heubeck, Formal analysis, Investigation, Visualization; Morgan DA Weiss, Investigation, Writing - review and editing; Xiao-jun Li, Methodology, Writing - review and editing; Adam K Savage, Methodology; Richard R Green, Data curation; Troy R Torgerson, Supervision, Writing - review and editing; Thomas F Bumol, Supervision, Investigation, Project administration, Writing - review and editing; Lucas T Graybuck, Conceptualization, Software, Formal analysis, Investigation, Visualization, Methodology, Writing - original draft, Writing - review and editing; Peter J Skene, Conceptualization, Supervision, Funding acquisition, Writing - original draft, Project administration, Writing - review and editing

### Author ORCIDs
Elliott Swanson http://orcid.org/0000-0002-0351-6446
Cara Lord http://orcid.org/0000-0002-7827-2159
Julian Reading http://orcid.org/0000-0002-8533-3992
Alexander T Heubeck http://orcid.org/0000-0002-9312-4090
Palak C Genge https://orcid.org/0000-0002-0851-6445
Zachary Thomson https://orcid.org/0000-0002-3861-9998
Morgan DA Weiss https://orcid.org/0000-0002-4189-1992
Xiao-jun Li http://orcid.org/0000-0003-0832-9479
Adam K Savage http://orcid.org/0000-0003-1913-5485
Richard R Green https://orcid.org/0000-0001-8851-7204
Troy R Torgerson https://orcid.org/0000-0003-3489-5036
Thomas F Bumol https://orcid.org/0000-0003-4319-1006
Lucas T Graybuck https://orcid.org/0000-0002-8814-6818
Peter J Skene https://orcid.org/0000-0001-8965-5326

### Ethics
Human subjects: Biological specimens purchased from BioIVT were collected under BioIVT Protocol SERATRIALS-18002 (WIRB Protocol #20190318), and were provided as cryopreserved PBMCs. Specimens purchased from Bloodworks NW were collected under BloodWorks Protocol BT001 (WIRB Protocol #20141589) as freshly drawn whole blood. All sample collections were conducted by BioIVT and Bloodworks NW under these IRB protocols, and all donors signed informed consent forms.

### Decision letter and Author response
Decision letter https://doi.org/10.7554/eLife.63632.sa1
Author response https://doi.org/10.7554/eLife.63632.sa2

# Additional files

## Supplementary files

• Supplementary file 1. Antibody-fluorophore conjugates used to characterize peripheral blood mononuclear cell cell-type abundance by flow cytometry before and after anti-CD15 bead-based neutrophil depletion.

• Supplementary file 2. Quantification of major cell populations by flow cytometry before (pre-depletion) and after (post-depletion) anti-CD15 bead-based neutrophil depletion of either Ficoll-purified or leukapheresis-purified peripheral blood mononuclear cells. The gating strategy used to assess these populations is presented in *Figure 1—figure supplement 4*.

• Supplementary file 3. Antibody-fluorophore conjugates used to assess peripheral blood mononuclear cell cell-type populations for evaluation of cell-type labeling.

• Supplementary file 4. Quantification of peripheral blood mononuclear cell cell-type populations using the antibody panel provided in *Supplementary file 3*. The gating strategy used to assess these populations is presented in *Figure 2—figure supplement 1*. Type labels and proportions used for comparisons of scATAC-seq cell-type labeling in *Figure 2d* are provided in the last two columns. When two gated populations are combined to tabulate a single cell type, the same cell-type label is listed beside each gate. Cell-type proportions are listed only at the first instance of each cell type.

• Supplementary file 5. Custom oligonucleotide sequences used to generate ICICLE-seq libraries.

• Supplementary file 6. Antibody-oligo conjugates (BioLegend TotalSeq-A) used for ICICLE-seq and TEA-seq experiments to label peripheral blood mononuclear cell cell types.

• Supplementary file 7. Sources and presentation of human biological specimens utilized in this study.

• Transparent reporting form

## Data availability

Raw scATAC-seq and ICICLE-seq, and TEA-seq data are deposited dbGaP for controlled access (dbGaP atudy accession phs002316.v1.p1). Processed data has been deposited in GEO (accession number GSE158013).

The following dataset was generated:

| Author(s) | Year | Dataset title | Dataset URL | Database and Identifier |
|---|---|---|---|---|
| Swanson E, Lord C, Green RR, Bumol TF, Graybuck LT, Skene PJ | 2020 | Integrated single cell analysis of chromatin accessibility and cell surface markers | https://www.ncbi.nlm.nih.gov/geo/query/acc.cgi?acc=GSE158013 | NCBI Gene Expression Omnibus, GSE158013 |

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
