## [Decision Letter]

**Acceptance summary:**

Swanson et al. present TEA-seq that enables simultaneous profiling of chromatin accessibility, RNA, protein epitope profiling from the same individual cells. This method uses an optimized lysis protocol that retains cellular membranes but allows for the capture of high quality chromatin accessibility data. TEA-seq has been optimized and shown to work in blood cells.

**Decision letter after peer review:**

Thank you for submitting your article "TEA-seq: a trimodal assay for integrated single cell measurement of transcripts, epitopes, and chromatin accessibility" for consideration by *eLife*. Your article has been reviewed by three peer reviewers, and the evaluation has been overseen by a Reviewing Editor and Jessica Tyler as the Senior Editor. The following individual involved in review of your submission has agreed to reveal their identity: Andrew C Adey (Reviewer #3).

The reviewers have discussed the reviews with one another and the Reviewing Editor has drafted this decision to help you prepare a revised submission.

We would like to draw your attention to changes in our policy on revisions we have made in response to COVID-19 (https://elifesciences.org/articles/57162). Specifically, when editors judge that a submitted work as a whole belongs in *eLife* but that some conclusions require a modest amount of additional new data, as they do with your paper, we are asking that the manuscript be revised to either limit claims to those supported by data in hand, or to explicitly state that the relevant conclusions require additional supporting data.

Summary:

Swanson et al. present two novel methods, ICICLE-seq and TEA-seq, which both benefit from an optimized lysis protocol that retains cellular membranes but allows for the capture of high quality chromatin accessibility data. This enables the simultaneous profiling of chromatin accessibility and cellular epitopes (ICICLE-seq) and can be used in conjunction with the 10x Genomics Multiome kit to additionally acquire cellular epitopes. The strengths of the manuscript are the attention to detail, the clarity of the methods, and the availability of all data. The primary weakness is in the presentation of ICICLE-seq and TEA-seq as related methods and in the lack of analytical exploration of the TEA-seq data.

Essential revisions:

1) The differences and similarities between ICICLE-seq and TEA-seq are confusing and their presentation in a single paper seems convoluted. These are two different methods that are enabled by a specific lysis protocol. Given that the title exclusively focuses on TEA-seq but TEA-seq occupies a very small percentage of the manuscript's real estate and innovation, we think the manuscript could benefit from a clearer presentation of TEA-seq.

2) Extension of TEA-seq to samples beyond PBMC. The reported optimizations are highly specific to PBMCs and this could be made clearer. There are many reasons to think that these same optimizations might not hold up in other cell types. The authors should either (a) perform the same comparisons on different cell lines or cell types that have proven to be troublesome in previous ATAC-seq experiments or (b) at least make it more explicit that these optimizations may not hold for other cell types. A good example of the former is K562 cells which had very high mitochondrial DNA contamination and very low signal to noise in the original ATAC-seq protocol.

3) Deeper analysis leveraging the 3-way single cell data in TEA-seq. Some additional analysis as to the improvement of separation and identification of cell types using the 3-way manifold should be explored. i.e. perform clustering and cell type ID on each independently and then identify overlap / which cell types are poorly separated in one modality but not another and then how the 3-way manifold performs as a comparison. Overall it looks like the RNA performs fine for cell type ID when compared to the 3-way with the other modalities performing worse (though that is form UMAP visualizations which are not able to be quantitatively interpreted for these purposes). In addition,, given the tri-modal data derived from TEA-seq, the authors should have the ability to assess how well scATAC-seq or scRNA-seq correlate with protein levels, albeit for a small set of proteins biased for cell surface proteins.

---

## [Author Response]

Essential revisions:1) The differences and similarities between ICICLE-seq and TEA-seq are confusing and their presentation in a single paper seems convoluted. These are two different methods that are enabled by a specific lysis protocol. Given that the title exclusively focuses on TEA-seq but TEA-seq occupies a very small percentage of the manuscript's real estate and innovation, we think the manuscript could benefit from a clearer presentation of TEA-seq.

We thank the reviewers for this constructive feedback, and agree that the framing and emphasis of the manuscript were imbalanced. We have updated the Abstract, Introduction, and Results sections to better frame the focus of the manuscript on TEA-seq, and demonstrate the role of ICICLE-seq as part of the progression towards the trimodal assay, rather than a truly standalone method. To better emphasize TEA-seq, we have expanded both the experimental and analytical results related to TEA-seq by the inclusion of a direct comparison between TEA-seq, CITE-seq, and scATAC-seq on the same set of samples, now presented as Figure 4—figure supplement 4.

Because ICICLE-seq was the first proof of concept that we could co-capture surface protein and nuclear chromatin state information (not demonstrated in the literature prior to the nearly simultaneous release of our preprint and the ASAP-seq preprint by Mimitou, Leareau, and Chen, et al. on bioRxiv), and our approach to this co-assay using poly-A tagged Tn5 complexes remains novel, we feel that ICICLE-seq still warrants placement within this article. However, we have worked to streamline the presentation of both the scATAC-seq improvements and ICICLE-seq results with the following modifications:

a) Figure 2 has been divided into a smaller main figure, retaining panels A-D, and an additional figure supplement, containing panels E-F, to streamline the presentation of scATAC-seq results.

b) Results and Figure 3 for ICICLE-seq data have been reduced and refocused to better emphasize the role of ICICLE-seq as a proof-of-concept for simultaneous capture of chromatin and cell surface epitopes. We have removed panels F and G from Figure 3 to avoid overreaching the results on this limited dataset.

c) We have updated the text to emphasize the role of ICICLE-seq as a proof of concept, rather than a standalone assay.

2) Extension of TEA-seq to samples beyond PBMC. The reported optimizations are highly specific to PBMCs and this could be made clearer. There are many reasons to think that these same optimizations might not hold up in other cell types. The authors should either (a) perform the same comparisons on different cell lines or cell types that have proven to be troublesome in previous ATAC-seq experiments or (b) at least make it more explicit that these optimizations may not hold for other cell types. A good example of the former is K562 cells which had very high mitochondrial DNA contamination and very low signal to noise in the original ATAC-seq protocol.

We agree that TEA-seq will have applications beyond PBMC samples, and that we have not demonstrated the applicability of our methods to additional cell and tissue types.

We have emphasized the focus of these methodological improvements on immune cell samples throughout the manuscript. The Introduction and Discussion sections both begin with a description of the use of PBMCs as critical, medically relevant samples, and PBMCs are the only cell type used throughout the manuscript. A critical step in our scATAC-seq improvements is the removal of neutrophils, which may not be present in other tissues, except in the case of tissue invasion in response to an insult.

To address this, we have expanded the Discussion section to provide a prospectus on the use of TEA-seq and the applicability of the scATAC-seq methods improvements we have presented in the manuscript to other cell types and tissues:

"It is likely that the use of TEA-seq in other cell types will require sample-specific optimization. […] This can be readily tested using bulk ATAC-seq and by measuring library complexity, FRIP score and mitochondrial contamination to determine data quality."

We hope that the additional discussion of applications and caveats for other cell types and tissues are helpful to the reviewers and readers.

3) Deeper analysis leveraging the 3-way single cell data in TEA-seq. Some additional analysis as to the improvement of separation and identification of cell types using the 3-way manifold should be explored. i.e. perform clustering and cell type ID on each independently and then identify overlap / which cell types are poorly separated in one modality but not another and then how the 3-way manifold performs as a comparison. Overall it looks like the RNA performs fine for cell type ID when compared to the 3-way with the other modalities performing worse (though that is form UMAP visualizations which are not able to be quantitatively interpreted for these purposes). In addition,, given the tri-modal data derived from TEA-seq, the authors should have the ability to assess how well scATAC-seq or scRNA-seq correlate with protein levels, albeit for a small set of proteins biased for cell surface proteins.

Deeper analysis of TEA-seq, as a central experiment in this manuscript, is certainly a welcome addition. To address this feedback, we have added additional TEA-seq experiments across multiple wells, and have performed additional analysis of the data with a focus on identifying putative regulatory elements that would be missed through the use of a paired RNA-ATAC assay alone:

a) We utilized the ArchR package to perform Peak-To-Protein linkage analysis, now described in the new Results section titled "Expanding cis-regulatory module identification with TEA-seq." In addition to panels that compare correlation scores using peaks vs. genes for each epitope in our TEA-seq antibody panel, we include in-depth views of an expanded cis-regulatory repertoire identified for two targets: CD192/CCR2 and CD38 in new panels Figure 4G-J. This new analysis shows the strength of linked measurements – one is able to take advantage of measurement of both protein and RNA to identify putative regulatory regions that may be critical for cell function.

b) Using aliquots of exactly the same PBMC sample vial as the expanded TEA-seq dataset, we performed whole-cell CITE-seq, permeabilized cell scATAC-seq, and 10x Multiome (RNA + ATAC) sequencing (both for purified nuclei and permeabilized cells) to provide a comprehensive set of comparisons to previous methodology. This expanded set of direct methodological comparisons allows us to present the strengths and weaknesses of the TEA-seq protocol, including better insight into the loss of cytosolic RNAs. We provide these comparisons in a new Figure 4—figure supplement 5 that demonstrates that TEA-seq ATAC and ADT data quality remain high after permeabilization and the RNA data is most directly comparable to single nucleus preparations.

We hope that these major new additions to the manuscript will be satisfactory to address the reviewers' comments, and that reviewers will support the publication and release of these comprehensive comparative datasets, which should allow analysis-focused researchers to develop more comprehensive analytical approaches for trimodal directly linked datasets.

In the future, we hope to provide a Research Advance to investigate cell hashing and methods to improve cytoplasmic RNA retention as we continue to optimize and develop methods around the core platform of TEA-seq. However, these advances and optimizations will take additional time as we work around COVID-19 restrictions.